# Intercomparison of four airborne imaging DOAS systems for tropospheric NO$_2$ mapping - The AROMAPEX campaign

Frederik Tack[1], Alexis Merlaud[1], Andreas C. Meier[2], Tim Vlemmix[3,a], Thomas Ruhtz[4], Marian-Daniel Iordache[5], Xinrui Ge[3,b], Len van der Wal[6], Dirk Schuettemeyer[7], Magdalena Ardelean[8], Andreea Calcan[8], Daniel Constantin[9], Anja Schönhardt[2], Koen Meuleman[5], Andreas Richter[2], and Michel Van Roozendael[1]

[1] BIRA-IASB, Royal Belgian Institute for Space Aeronomy, Brussels, Belgium
[2] IUP-Bremen, Institute of Environmental Physics, University of Bremen, Germany
[3] TU Delft, Delft University of Technology, Delft, The Netherlands
[4] FUB, Institute for Space Sciences, Freie Universität Berlin, Germany
[5] VITO-TAP, Flemish Institute for Technological Research, Mol, Belgium
[6] TNO, Netherlands Organisation for Applied Scientific Research, The Netherlands
[7] ESA-ESTEC, European Space Agency, Noordwijk, The Netherlands
[8] INCAS, National Institute for Aerospace Research "Elie Carafoli", Bucharest, Romania
[9] "Dunarea de Jos" University of Galati, Galati, Romania
[a] now at: KNMI, Royal Netherlands Meteorological Institute, De Bilt, The Netherlands
[b] now at: WUR, Wageningen University and Research, The Netherlands

*Correspondence to:* Frederik Tack (frederik.tack@aeronomie.be)

**Abstract.** We present an intercomparison study of four airborne imaging DOAS instruments, dedicated to the retrieval and high resolution mapping of tropospheric nitrogen dioxide (NO$_2$) vertical column densities (VCDs). The AROMAPEX campaign took place in Berlin, Germany in April, 2016 with the primary objective to test and intercompare the performance of experimental airborne imagers. The imaging DOAS instruments were operated simultaneously from two manned aircraft, performing synchronised flights: APEX (VITO/BIRA-IASB) was operated from DLR's DO-228 D-CFFU aircraft at 6.2 km altitude, while AirMAP (IUP-Bremen), SWING (BIRA-IASB) and SBI (TNO/TU Delft/KNMI) were operated from the FUB Cessna 207T D-EAFU at 3.1 km. Two synchronised flights took place on 21 April 2016. NO$_2$ slant columns were retrieved by applying differential optical absorption spectroscopy (DOAS) in the visible wavelength region and converted to VCDs by the computation of appropriate air mass factors (AMFs). Finally, the NO$_2$ VCDs were georeferenced and mapped at high spatial resolution. For the sake of harmonising the different data sets, efforts were made to agree on a common set of parameter settings, AMF LUT and gridding algorithm. The NO$_2$ horizontal distribution, observed by the different DOAS imagers, shows very similar spatial patterns. The NO$_2$ field is dominated by two large plumes related to industrial compounds, crossing the city from west to east. The major highways A100 and A113 are also identified as line sources of NO$_2$. Retrieved NO$_2$ VCDs range between 1 x 10$^{15}$ molec cm$^{-2}$ upwind of the city and 20 x 10$^{15}$ molec cm$^{-2}$ in the dominant plume, with a mean of 7.3 ± 1.8 x 10$^{15}$ molec cm$^{-2}$ for the morning flight and between 1 and 23 x 10$^{15}$ molec cm$^{-2}$ with a mean of 6.0 ± 1.4 x 10$^{15}$ molec cm$^{-2}$ for the afternoon flight. The mean NO$_2$ VCD retrieval errors are in the range of 22 to 36

% for all sensors. The four data sets are in good agreement with Pearson correlation coefficients better than 0.9, while the linear regression analyses show slopes close to unity and generally small intercepts.

## 1 Introduction

Currently, almost 60 % of the world population is living in urban areas, where they are exposed to emissions from the majority of anthropogenically produced air pollutants. Nitrogen dioxide ($NO_2$) is a trace gas and key pollutant that can be considered as a proxy for air quality/pollution in an urban environment, as it mainly originates from combustion processes such as burning of fossil fuels which are mainly related to traffic and industry. $NO_2$ plays an important role in atmospheric chemistry and can have a direct impact on human health. It is a short-lived species with a strong local character and concentrations that can vary strongly both in space and time. For the reasons stated, the monitoring and high resolution mapping of the $NO_2$ distribution is considered to be of great (social) relevance.

For about two decades, tropospheric trace gases, such as $NO_2$, have been monitored and mapped at a global scale by spaceborne sensors like ESA's SCIAMACHY (SCanning Imaging Absorption spectroMeter for Atmospheric CHartographY), ESA's GOME (Global Ozone Monitoring Experiment), ESA's/EUMETSAT's GOME-2, and NASA's OMI (Ozone Monitoring Instrument). See for example Richter and Burrows (2002), Beirle et al. (2010), Boersma et al. (2011), Hilboll et al. (2013), Valks et al. (2011) and Bucsela et al. (2013). However, the coarse spatial resolution in the order of a few tens of kilometers of these spaceborne air quality instruments makes them ineffective for studies of the $NO_2$ field at the scale of cities and for resolving individual emission sources.

In the last decade, a number of studies have explored the potential of airborne imaging DOAS systems for higher-resolution mapping of the spatial distribution of tropospheric gases. The majority of these studies have focused on the retrieval of the $NO_2$ field over urban areas and/or industrial sites, i.e. Heue et al. (2008), Kowalewski and Janz (2009), Popp et al. (2012), General et al. (2014), Lawrence et al. (2015), Schönhardt et al. (2015), Nowlan et al. (2016), Lamsal et al. (2017), Meier et al. (2017), Tack et al. (2017), Vlemmix et al. (2017), Broccardo et al. (2018), Merlaud et al. (2018), and Nowlan et al. (2018).

As the developed instruments vary in design, size, specifications and data analysis applied, it is interesting to compare results from simultaneous observations. Here we present the first intercomparison study of $NO_2$ VCDs, retrieved by the differential optical absorption spectroscopy (DOAS) analysis of visible spectra, observed by four different airborne imaging DOAS spectrometers. The instruments were operated simultaneously from two manned aircraft over Berlin during the ESA funded AROMAPEX campaign taking place in April 2016.

The primary objective of the AROMAPEX project was to test and intercompare experimental airborne atmospheric imagers, dedicated to the geographical mapping of the spatial distribution of tropospheric $NO_2$. AROMAPEX is also a preparatory step for forthcoming intercomparison/validation campaigns of satellite air quality sensors. In the coming years, a new generation of spaceborne instruments will be launched, providing information on atmospheric variables at much higher

spatial resolution, in the order of a few kilometers. These measurements will be valuable for air quality, atmospheric composition and climate monitoring studies/services. ESA has launched Sentinel-5 Precursor (S-5P) on 13 October 2017, being a sun-synchronous low earth orbit (LEO) mission (Ingmann et al., 2012), and has planned the launch of the first Sentinel-5 (S-5) in 2021. Additionally, a range of geostationary (GEO) missions are planned: ESA's Sentinel-4 (S-4) (Ingmann et al., 2012), NASA's TEMPO (Tropospheric Emissions: Monitoring of POllution; Chance et al., 2013; Zoogman et al., 2017) and KARI's GEMS (Geostationary Environmental Monitoring Spectrometer; Kim, 2012). The unprecedented characteristics of these instruments, such as higher spatial and temporal resolution will create many new science opportunities, but also retrieval challenges. The AROMAPEX campaign and study are aimed at the preparation of the validation of trace gas products from future spaceborne systems and for the study of satellite intra-pixel variability.

The manuscript is organised as follows: Sect. 2 presents the context of the AROMAPEX project and provides details about the set-up of the airborne campaign held in Berlin. Sect. 3 briefly introduces the four airborne imaging DOAS systems, operated during AROMAPEX. Sect. 4 describes the data analysis of the airborne observations for the retrieval and geographical mapping of the $NO_2$ VCDs. In the following two sections, the resulting $NO_2$ VCD distribution maps are discussed and compared with mobile car-DOAS measurements. Sect. 7 discusses a quantitative assessment by intercomparing the co-located $NO_2$ VCD products, retrieved from the four imagers.

## 2 The AROMAPEX campaign

The AROMAPEX campaign was held in Berlin from 11 April to 22 April, 2016. An overview of the area, flight plan and main campaign sites is provided in Fig. 1. The four imaging DOAS systems were operated from two manned aircraft, performing time-synchronised flights at different altitudes: APEX (Airborne Prism Experiment) was operated from the DO-228 D-CFFU aircraft of DLR (Deutsches Zentrum für Luft- und Raumfahrt) at 6.2 km a.g.l., while AirMAP (Airborne imaging DOAS instrument for Measurements of Atmospheric Pollution), SWING (Small Whiskbroom Imager for trace gases monitoriNG) and Spectrolite Breadboard Instrument (SBI) were operated from the Cessna 207T D-EAFU of FUB (Free University Berlin) at 3.1 km a.g.l. The cruise altitudes of both aircraft were well above the planetary boundary layer (PBL), containing the majority of tropospheric $NO_2$. The aircraft operated from the Schönhagen airfield (see Fig. 1), 40 km south-west of Berlin, while the research teams were based at the Institute for Space Sciences of FUB, where measurements of additional atmospheric parameters were made.

The complex flight constellation was carefully planned in order to optimise the acquisition for trace gas retrieval purposes. Due to rainy and cloudy weather conditions at the beginning of the campaign, the two scheduled flights both took place on 21 April, the only clear-sky day during the campaign (see Table 1). The first flight took place in the morning from 09:34 to 12:01 LT and the second flight in the afternoon from 14:24 to 16:39 LT. The entire city of Berlin was covered by both flights, as well as the semi-urban and rural area east and south of the city. An area of approximately 800 km$^2$ was covered, consisting of 14 flight lines for the morning and afternoon flight. Note that due to a small delay of the Dornier

aircraft, the second flight line of the morning flight was skipped in order to be better time-synchronised with the Cessna. This explains the data gap in the retrieved APEX $NO_2$ VCD distribution map (see Fig. 11). The absolute temporal offset between both aircraft above a certain position was 10 and 12 minutes on average for the morning and afternoon flight, respectively, with a maximum time difference of 24 minutes.

The flight plan consisted of adjacent straight flight lines, alternately flown from south to north, and from north to south, with the first flight line in the west. Due to the large roll angles, spectra acquired during turns of the aircraft in between flight lines are not taken into account in the comparison. The flight plan approved by air traffic control (ATC) was initially larger than the area covered, in order to have some flexibility to adapt the actual flight pattern to the wind direction. Downwind of the sources, a maximum number of flight lines were retained in order to catch the urban plume. Upwind of the

main known sources, the number of flight lines were reduced. In the case of the flights on 21 April, more flight lines were foreseen in the east as a result of the predicted west wind.

The AROMAPEX campaign is part of the AROMAT-I and –II (Airborne Romanian Measurements of Aerosols and Trace gases) activities (Constantin and Merlaud, 2016), held in Romania in September, 2014 and August, 2015. The campaign was initially planned to take place in Romania, Bucharest in summer 2015, but was eventually rescheduled to take

place over Berlin in spring 2016, due to critical issues with the flight approvals over Romania for the DLR Dornier aircraft. AROMAPEX builds on the experience gained during the AROMAT campaigns (first flights with AirMAP and SWING together for $NO_2$ and $SO_2$ retrievals) and the BUMBA campaigns (Belgian Urban $NO_2$ Monitoring Based on APEX remote sensing) held in April-June, 2015 and July, 2016 in Belgium (Tack et al., 2017).

## 3 Airborne imaging DOAS instruments and data sets

The characteristics of the four airborne imaging DOAS instruments, which were operated during the AROMAPEX campaign, are only briefly discussed here with a focus on their differences, and the main specifications are summarised in Table 2. References are provided below, containing a more detailed and technical discussion of each instrument and data analysis. A mosaic of the four imaging instruments is shown in Fig. 2.

### 3.1 APEX

Airborne Prism EXperiment (APEX) is a pushbroom imaging spectrometer developed by a Swiss-Belgian consortium (the Flemish Institute for Technological Research (VITO) and the Remote Sensing Laboratories (RSL) at the Department of Geography of the University of Zürich) on behalf of ESA (Itten et al., 2008; D'Odorico, 2012; Schaepman et al., 2015). Although APEX is initially designed as an airborne remote sensing instrument for land use – land cover (LULC) applications, several studies have demonstrated that the instrument is suitable for atmospheric trace gas retrieval

applications, and in particular $NO_2$ (Popp et al., 2012; Kuhlmann et al., 2016; Tack et al., 2017). APEX records data in the visible, near infra-red and infrared regions of the electromagnetic spectrum, covering the wavelength range between 370 and

2540 nm. The radiance is spectrally dispersed by a prism, while the three other imaging instruments are equipped with a grating spectrograph. Because of the use of a prism dispersion element, the full width at half maximum (FWHM) is a non-linear function, broadening with wavelength. In the visible wavelength range, the spectral resolution increases from 1.5 to 3 nm FWHM. APEX has an across-track field of view (FOV) of 28˚ and records data in 1000 across-track pixels. A swath width of 3.1 km is obtained at a typical flight altitude of 6.2 km a.g.l. In order to obtain a favorable signal-to-noise (SNR) for trace gas retrieval, spectra are spatially binned by 20 pixels along- and across-track resulting in a spatial resolution of approximately 80 by 60 $m^2$. The native detection limit with respect to $NO_2$ DSCD retrievals is ~$3.3 \times 10^{15}$ molec $cm^{-2}$. Note that the spatial resolution is considerably higher than the typical resolution of spaceborne sensors for the monitoring of the atmospheric composition: one OMI pixel of 13 by 24 $km^2$ and one TROPOMI (TROPOspheric Monitoring Instrument) pixel of 3.5 by 7 $km^2$ are covered by approximately 65000 and 5000 APEX pixels, respectively. The latter is the spectrometer payload of the ESA Sentinel-5 Precursor satellite, launched in October 2017. The APEX optical unit is enclosed by a thermo-regulated box in order to be temperature stabilised, while the pressure in the spectrometer is kept at 200 hPa above ambient pressure.

### 3.2 AirMAP

The Airborne imaging DOAS instrument for Measurements of Atmospheric Pollution (AirMAP) has been developed for the purpose of airborne trace gas measurements and pollution mapping by the Institute of Environmental Physics in Bremen (IUP-Bremen). The instrument specifications and previous campaign results have been thoroughly discussed in Schönhardt et al. (2015) and Meier et al. (2017). AirMAP is a pushbroom UV-Vis imager with a wide FOV of around 51.7°, resulting in a swath width of approximately the same size as the flight altitude. The wavelength region and spectral resolution can be customised according to the chemical species of interest, with a spectral coverage of either 41, 63 or 86 nm, depending on the grating used. For the AROMAPEX campaign, AirMAP was equipped with a 400 g/mm grating blazed at 400 nm, enabling measurements of the incoming light in the 429-492 nm wavelength range, with a spectral resolution between 0.9 and 1.6 nm FWHM. From a maximum of 35 individual lines of sight (LOS), represented by 35 single fibers, the number of viewing directions is adapted to each situation by averaging according to SNR or spatial resolution requirements. The spectra acquired during AROMAPEX have a spatial resolution of approximately 30 m along-track and 86 m across-track and the approximate detection limit with respect to $NO_2$ DSCD retrievals is ~$2.2 \times 10^{15}$ molec $cm^{-2}$. The spectrometer is temperature stabilised at 35 °C.

### 3.3 SWING v2

The Small Whiskbroom Imager for atmospheric composition monitoriNG (SWING) was developed by the Royal Belgian Institute for Space Aeronomy (BIRA-IASB) based on the experience, gained with previous (airborne) DOAS instruments (Merlaud et al., 2011; Merlaud et al., 2012). The compact payload is initially designed to be operated from an Unmanned Aerial Vehicle (UAV) and first results of this instrumental set-up were discussed in Merlaud et al. (2013, 2018). During the

AROMAPEX campaign, an upgraded version of SWING was operated from the FUB Cessna alongside AirMAP and SBI. SWING v2 was deployed for the first time during the AROMAT-2 campaign, in order to measure $NO_2$ and $SO_2$ in the exhaust plume of a Romanian power plant (Constantin and Merlaud, 2016). SWING v2 is based on an AVANTES AvaSpec-ULS2048-XL UV-Vis spectrometer covering the wavelength range 280 - 550 nm at a spectral resolution of 0.7 nm FWHM.

A PC-104 (Lippert CSR LX800) runs the acquisition software and stores the acquired spectra. Scattered solar radiation from different LOS is collected by a rotating mirror which is mounted on a HITEC HS-5056-MG servomotor, controlled by an Arduino Micro. The mirror is able to scan at a maximum FOV of 110°, but was tuned to a FOV of 50° for the AROMAPEX campaign in order to yield a similar swath width as AirMAP. In contrast to APEX and AirMAP, SWING is a lightweight, compact whiskbroom instrument. Including the housing and the electronics, the weight, size, and power consumption of

SWING are respectively 1200 g, 33x12x8 cm$^3$ and 10 W. The main reason for implementing a whiskbroom set-up were the constraints both in weight and size, in order to be operated from an UAV. A disadvantage of this instrumental set-up is, however, that $NO_2$ maps are not built continuously by consecutive scan lines but by a cloud of scanned points. In the AROMAPEX flight geometry, the SWING large instantaneous field of view (IFOV) of 6° yielded continuous maps with a spatial resolution of approximately 325 m and a DSCD detection limit of $\sim 1.8 \times 10^{15}$ molec cm$^{-2}$. From the perspective of

the analysis, a whiskbroom set-up has the advantage that it requires only one calibration set in the DOAS analysis, instead of a calibration set per across-track detector.

## 3.4 SBI

The Spectrolite Breadboard Instrument (SBI) is a compact UV-Vis pushbroom spectrometer that has been developed at the Netherlands Organisation for Applied Scientific Research (TNO) for various applications (air quality, land use, water quality

monitoring). The instrument is designed to operate from a 12-Unit CubeSat and its size and weight are 31x42x19 cm$^3$ and 8 kg, respectively. Although primarily designed for future application in space, SBI was adapted to an airborne instrument and performed its maiden flight during the AROMAPEX campaign. The instrument specifications are discussed in more details in de Goeij et al. (2016), while the $NO_2$ retrieval approach and AROMAPEX campaign results are reported in Ge and Vlemmix (2016), and Vlemmix et al. (2017). It was decided only shortly before the AROMAPEX campaign to add SBI to

the instrumental set-up, which made it an ambitious and challenging task to get the breadboard ready. Due to technical reasons, a temporary, but non-optimal, telescope was used with a narrow FOV of 8.3°. This limited the swath width to 450 m at a flight altitude of 3.1 km a.g.l., which is considerably smaller than for the other imagers. SBI has a spectral coverage from 320 to 500 nm with a spectral resolution of 0.3 nm FWHM. However, other spectral ranges are possible between 270 and 2400 nm without affecting the design. Spectra were only binned in the along-track direction, resulting in a spatial resolution

of approximately 6 by 205 m$^2$ and an approximate DSCD detection limit of $\sim 2.2 \times 10^{15}$ molec cm$^{-2}$. The instrument is stabilised at a temperature of 25 °C.

# 4 Retrieval of NO₂ vertical column densities

The retrieval and geographical mapping of NO$_2$ VCDs, based on spectra acquired by the airborne imagers, consists of a three-step approach. First, the well-established DOAS technique (Platt and Stutz, 2008), based on the Beer-Lambert law, is applied on the observed backscattered solar radiation in the visible wavelength region (Sect 4.1). For each analysed spectrum, this results in the retrieval of a slant column density (SCD), being the concentration of NO$_2$ integrated along the effective viewing path. SCDs depend on the optical path of the observation and are thus strongly dependent on the viewing geometry and the radiative transfer. In the next step, an air mass factor (AMF; Solomon et al., 1987) is computed for each observation by modeling an assumed state of the atmosphere and transfer of the solar radiation through the atmosphere, based on a radiative transfer model (RTM) (Sect 4.2). AMFs are the factor between the slant and the vertical column, accounting for the effects of viewing and sun geometry, surface reflectance, aerosol scattering and the NO$_2$ vertical distribution. SCDs from the DOAS fit can then be converted to VCDs, being the integrated amount of NO$_2$ along a single vertical transect from the Earth's surface to the top of the atmosphere:

$$VCD_i = \frac{SCD_i}{AMF_i} \qquad (1)$$

VCDs are a more geophysical relevant quantity, independent of changes in the optical path length of the SCDs, e.g. due to high surface reflectance or large solar zenith angle. VCD retrievals from different DOAS instruments can therefore be compared in a meaningful way. In a third and final step, the observations are combined with the recorded sensor position and orientation, allowing a proper geographical mapping of the NO$_2$ VCDs (Sect 4.3). The retrieval approaches and (the impact of) the parameter settings are only briefly discussed in the next sections. For full details on the APEX, AirMAP, SWING and SBI retrieval approaches, we refer respectively to Tack et al. (2017), Meier et al. (2017), Merlaud et al. (2018), and Vlemmix et al. (2017).

## 4.1 DOAS analysis of the observed spectra

A DOAS analysis was applied first to all the observed spectra in order to retrieve NO$_2$ slant columns. The DOAS approach separates the broadband (Earth's surface reflectance, Rayleigh and Mie scattering) and narrow band (molecular absorption) signals in the observed spectra by fitting a low-order polynomial term and isolating the rapidly varying molecular absorption structures. Then, absorption cross-sections of NO$_2$ and interfering trace gases, such as O$_3$, O$_4$ and H$_2$O, and a synthetic Ring spectrum are simultaneously fitted. The fitting interval was within 425 and 510 nm for all imagers. NO$_2$ exhibits strong spectral absorption structures in this region, while there is relatively low interference from absorption features of other trace gases. As the DOAS analysis parameters are largely dependent on the instrument, each involved group applied its own spectral fitting tool and optimised settings for NO$_2$ retrieval. The impact of using different DOAS retrieval tools has been studied in Peters et al. (2017) and an excellent overall correlation was reported. For each instrument, the main DOAS

analysis parameters and fitted absorption cross-sections are provided in Table 3. Note that $O_3$ and $H_2O$ cross-sections were not fitted in the APEX retrievals due to cross-correlations and overparameterisation of the small fitting interval. $O_4$ and $H_2O$ were not fitted in the SBI retrievals due to small absorption in the chosen fitting window. There were also no patterns visible in the residuals that correlated with the shape of the water vapour differential cross-section. This is also expected on such a clear-sky day over a relatively small region.

The direct output of the DOAS fit is not a SCD but a differential slant column density (DSCD), which is the integrated concentration of $NO_2$ along the effective light path with respect to the same quantity in a selected reference spectrum ($SCD_{ref}$). Reference spectra were acquired over a clean forest area, west (upwind) of the city center, characterized by a low and homogeneous $NO_2$ field and a low albedo variability. In case of a pushroom imager, a reference spectrum is required for each across-track detector, each having its intrinsic spectral response, in order to avoid across-track biases. For each flight, new reference spectra were acquired in order to reduce systematic biases due to changes in environmental conditions, affecting the instrument characteristics and its spectral performance. Several spectra were averaged in order to increase the SNR of the reference spectrum, e.g. in case of AirMAP 120 spectra were averaged over one minute, reducing the noise to approximately $2.0 \times 10^{14}$ molec $cm^{-2}$. It is assumed that the background spectrum contains a residual $NO_2$ amount of 1 x $10^{15}$ molec $cm^{-2}$. This value for the background correction is considered to be a typical value for an European summer month as shown in Huijnen et al. (2010). Due to the nature of the different instruments, a slightly different approach was applied for each instrument in order to acquire the reference spectrum. These have been extensively discussed in the related papers, reporting results from the individual involved airborne imagers (see Meier et al. (2017) for AirMAP, Tack et al. (2017) for APEX, Vlemmix et al. (2017) for SBI, and Merlaud et al. (2018) for SWING).

The differential approach (1) largely reduces the impact of systematic instabilities related to instrumental artefacts and the Fraunhofer lines, which blur out the much finer trace gas absorption features and (2) cancels out the stratospheric $NO_2$ contribution in the signal, assuming a small variability of the stratospheric $NO_2$ field between the acquisition of the analysed spectrum and the reference spectrum. Eq. (1) can be rewritten as:

$$VCD_i = \frac{DSCD_i + SCD_{ref}}{AMF_i} \tag{2}$$

Or

$$VCD_i = \frac{DSCD_i + \left(VCD_{ref} * AMF_{ref}\right)}{AMF_i} \tag{3}$$

Prior to the DOAS analysis, a spectral calibration was applied in order to obtain the instrument spectral response function (ISRF or slit function) as well as to accurately align the analysed spectrum, the reference spectrum and the absorption cross-sections in the DOAS fit. The accurate pixel-to-wavelength mapping is either done by aligning the Fraunhofer lines in the in-

flight spectra with a high resolution solar atlas (APEX, SWING, SBI) or by HgCd line lamp measurements on the ground (AirMAP). The main details of the wavelength calibration are provided as well in Table 3.

The NO$_2$ SCD time series of AirMAP, SWING and SBI, the three DOAS systems that were mounted on the FUB Cessna, are shown in Fig. 3 for the morning flight over Berlin on 21 April 2016. Due to the dependency of slant columns on the optical path and thus on the viewing geometry and the radiative transfer, only near-nadir SCDs were compared by averaging the observations of each across-track scan between -4° and +4° viewing zenith angle (VZA). Note that differences in the effective noise levels are partly caused by differences in the instrument IFOV, integration time and averaging of observations. This is further discussed in Sect. 4.4. As APEX was operated at a different time and altitude, its SCDs are not shown in the comparison. The flight lines were alternately flown from south to north, and from north to south, with the first flight line in the west. A major east-west oriented plume was discovered in the northern part of the acquired area, originating from the  power plant "Reuter West". Each peak corresponds to the crossing of the main plume. In Fig. 4, a zoom on the SCD time series is shown between 08:21 and 08:36 UTC. The first peak corresponds to the crossing of the plume when the Cessna was flying to the north. Then the aircraft turned to prepare the acquisition of the next flight line in southern direction and crossed the same plume a second time. The NO$_2$ SCDs are $11 \times 10^{15}$ molec cm$^{-2}$ on average and agree very well with an average difference of less than $1 \times 10^{15}$ molec cm$^{-2}$ and Pearson correlation coefficients better than 0.9. This points out the robustness of the applied DOAS retrieval tools.

## 4.2 Air mass factor computation

The DSCDs retrieved by the DOAS analysis do not only depend on the absorber profile, but also on the light path, affected by the observation geometry, atmospheric conditions and Earth's surface reflectance. The state of the atmosphere and radiative transfer through the atmosphere needs to be properly modelled, to calculate appropriate air mass factors (AMFs) which are needed to convert the retrieved DSCDs to VCDs. NO$_2$ AMFs have been computed using the RTM package UVspec/DISORT (Mayer and Kylling, 2005). DISORT numerically reproduces the atmospheric state and the radiative transfer based on a priori information on the parameters that affect the slant column light path. These are the surface reflectance, sun and viewing geometry, and atmospheric properties, such as cloud cover, pressure, temperature, absorber and aerosol vertical profiles.

### 4.2.1 RTM parameters

**(1)** Both APEX and SBI are radiometrically calibrated, thus an effective surface reflectance can be derived directly from the observed at-sensor radiances, provided that an atmospheric correction is applied. AirMAP and SWING, on the other hand, are not radiometrically calibrated. In Meier et al. (2016) an approach is presented to estimate surface reflectances from the AirMAP observed intensities, after scaling or vicarious calibration using a reference region with well-known surface reflectance taken from the ADAM database (Prunet et al., 2013). For the SWING data, surface reflectances were taken from the APEX albedo product. In all cases, a Lambertian surface was assumed. **(2)** Viewing geometry and solar position, defined

by the viewing zenith angle (VZA), solar zenith angle (SZA) and relative azimuth angle (RAA), can be directly extracted for each observation. **(3)** The presence of clouds can strongly affect the optical path and usually requires the need for a cloud retrieval scheme, e.g. for spaceborne retrievals. However, this could be neglected as all flights were performed under cloud-free conditions. **(4)** Since no accurate $NO_2$ profile shape information was available over the city, assumptions on the vertical distribution of $NO_2$ needed to be made. A box profile, with constant mixing ratio in the PBL, was assumed for the $NO_2$ vertical distribution. A PBL height of respectively 525 m and 1075 m was established for the morning and afternoon flight, based on observations performed with a Ceilometer CHM15k. The instrument was mounted on the rooftop of the FUB Institute for Space Sciences, located in the southwest of the city (52.46° N, 13.31° E, 80 m a.s.l.; see Fig. 1). **(5)** During the morning and afternoon flight a low aerosol optical thickness (AOT Level 1.5) of respectively 0.09 and 0.06 was measured by the CIMEL AERONET station (Holben et al., 1998) at the FUB. The AOT was averaged between 440 (middle of the SBI $NO_2$ fitting interval) and 490 nm (middle of the APEX $NO_2$ fitting interval). The measurement site was, however, located upwind of the main sources on 21 April, 2016 and was probably underestimating the AOT over the city. For the whole month of April 2016, an average AOT of 0.13 was measured between 440 and 490 nm at the FUB AERONET station. In order to compensate for the possible underestimation of the aerosol loading and related uncertainties due to the site location, a representative AOT of 0.15 and 0.10 was used in the RTM for the morning and afternoon flight, respectively.

These values are largely consistent with measurements performed with a Model 540 Microtops II handheld sun-photometer from Solar Lights (Porter et al., 2001), operated from a car which was driving through the city of Berlin during the aircraft overpasses on 21 April 2016. In Fig. 5, a time series of retrieved AOTs at 500 nm is shown in the upper panel and a map is provided in the lower panel. Two similar routes were followed in the morning and afternoon, starting from the FUB Institute for Space Sciences. The mean and median AOT are 0.21 and 0.16, respectively, and a number of elevated values can be observed, which are probably related to local sources or contamination by sub-visible cirrus clouds. The first and last observation in the time series were performed at the FUB Institute for Space Sciences, thus very close to the CIMEL AERONET station. Both for the morning and afternoon, the Microtops AOTs are higher than the CIMEL AOTs. Two possible reasons are currently under investigation: First, the AERONET station has a higher and less polluted position on the rooftop of the Institute for Space Sciences. Secondly, there might be a calibration issue for the Microtops, despite the fact that it was calibrated in 2015.

Aerosol extinction profiles (AEP) were supposed to be measured directly from the Cessna, based on the airborne spectrometer system FUBISS-ASA2 (Zieger et al., 2007). The instrument provides simultaneous measurements of the direct solar irradiance and the aureole radiance in two different solid angles. Due to restrictions imposed by air traffic control, soundings could eventually not be performed directly over or near the city but were performed over a rural area south of Berlin and on a long descent track ending close to the Polish border. As these profiles were not representative for the city of Berlin, aerosol extinction profiles were constructed from the AOT and PBL heights, measured by the FUB CIMEL and ceilometer during the respective flights, using an assumed profile shape. Both profiles include 75 % of the AOT in the well-

mixed PBL, where the extinction is set constant, while the remaining 25 % above the PBL exponentially decrease with altitude. For all extinction profiles a single-scattering albedo (SSA) of 0.93 was assumed (Dubovik et al., 2002).

In DISORT, the radiative transfer equation is solved in a pseudo-spherical, multiple scattering atmosphere using the discrete ordinate method. Simulations are performed for two different sensor altitudes, i.e. 3.1 km (Cessna 207T D-EAFU) and 6.2 km (Dornier DO-228 D-CFFU) a.g.l., and four different wavelengths, i.e. 440, 462, 464 and 490 nm. These wavelengths represent the middle of the $NO_2$ fitting windows of the four different DOAS imagers (see Table 3). For the sake of harmonising the different data sets, a common $NO_2$ AMF look-up table (LUT) was computed. An overview of the used grid for the different RTM parameters in the AMF LUT is provided in Table 4. For each retrieved slant column, an AMF was extracted from the LUT based on the viewing geometry, solar position and surface reflectance using linear interpolation. Based on Eq. (3), the slant columns can then be converted to the more geophysical relevant VCDs.

### 4.2.2 AMF dependence on RTM parameters

### 4.2.2.1 AMF dependence on the surface reflectance

A time series of near-nadir $NO_2$ AMFs is shown in Fig. 6 for the morning flight on 21 April, 2016. The corresponding surface reflectances, and viewing and sun geometries recorded by the AirMAP instrument are also provided in the plot, as well as the other RTM parameter settings. A strong dependence of the AMF on the surface reflectance can be observed, consistent with previous studies reported in Lawrence et al. (2015), Meier et al. (2017) and Tack et al. (2017). In the upper panel of Fig. 10 can be observed that the dependence is non-linear, especially below a surface reflectance of 0.2. When the surface is bright, a large fraction of the incident sunlight is reflected from the ground back to the imager and, thus, for an $NO_2$ profile peaking close to the ground, a larger $NO_2$ slant column is retrieved than in the case of a low surface reflectance, even when considering the same $NO_2$ profile, sampled below the aircraft. Consequently, the computed AMF should be relatively high in case of a bright surface albedo to account for the higher measurement sensitivity and to properly compensate for the larger slant column. Urban environments usually exhibit a very strong variability in surface reflectance and subsequently in the AMF. A slight overall increase of the AMF can be observed in the middle of the flight where spectra are acquired over the city and suburban area, characterised by a higher albedo. The areas covered by the first and last flight lines have a rather rural and forrested character, resulting in an overall lower albedo, and thus, lower AMF. The mean surface reflectance and AMF are 0.03 and 1.7, respectively, for the AirMAP observations.

The surface reflectance products of APEX and AirMAP have been compared for the afternoon flight. As an extensive surface reflectance intercomparison study is beyond the scope of this paper, we refer to Meier (2017) for further details. For the APEX surface reflectance product, an atmospheric correction was applied to the observed at-sensor radiances according to the methodology described in Sterckx et al. (2016). The atmospheric correction parameters were tuned to ensure a good matching of APEX spectra with co-located ground truth reflectances, measured during the campaign with an ASD FieldSpec-4 spectrometer (http://www.asdi.com/products-and-services/fieldspec-spectroradiometers/fieldspec-4-hi-res) over

different target surfaces. The surface reflectances retrieved from APEX spectra are calibrated at 500 nm and have a high spatial resolution of 4 by 3 m$^2$. Besides the APEX surface reflectance product in the spectral range of 490–500 nm, used for the APEX AMF computations and close to the middle of the APEX $NO_2$ fitting interval, a second product was derived by averaging APEX surface reflectances along the spectral dimension in the interval between 438–490 nm, corresponding to the AirMAP DOAS fit window and consequently the spectral range in which AirMAP's surface reflectance product is retrieved. A comparison was also done with the surface reflectance product of the Landsat 8 Operational Land Imager (OLI) spaceborne instrument (Barsi et al., 2014), based on an overpass on the same day at 11:56 LT. Band-1 was used, covering the spectral range from 435 to 451 nm and with a spatial resolution of about 30 m.

The quantitative comparison was performed by binning the different data sets on a regular grid with a cell size of 0.0010° (110 by 68 m$^2$) in order to avoid significant differences caused by different spatial resolution. Pearson correlation coefficients were 0.85, 0.92, and 0.92, and linear regression slopes 1.09, 1.14, and 1.47 for the comparison of the AirMAP surface reflectance product with the Landsat(435–451 nm), APEX(438–490 nm) and APEX(490–500 nm) product, respectively. Histograms for the different surface reflectance products are shown in Fig. 7 for the afternoon flight. The surface reflectances retrieved from AirMAP, APEX and Landsat 8 agree well, especially for the most frequent surface reflectances found in the covered area. The AirMAP surface reflectances have, however, a lower dynamic range. With exception of AirMAP, all sensors show a frequent occurrence of very small surface reflectances close to zero. This is mainly related to the assumptions made on the parameters in the atmospheric correction and is mostly pronounced above dark areas, e.g. the lake site in the east and the forest in the west of the covered area. Also very large values are not found in the AirMAP retrievals. This lower dynamic range is at least partially caused by the lower spatial resolution of AirMAP and spatial blur due to reduced imaging capabilities of the instrument in comparison to APEX and Landsat. This may explain the pronounced slopes in the correlation plots, because a strong weight is given to these extreme points in the regression. The histograms also clearly show that the surface reflectances from the different sensors are offset against each other. This offset is likely to be caused by a combination of the radiometric calibration and the reference spectra used for the calibration of the surface reflectances, as well as an overestimation of the path radiance, i.e. the radiance scattered in the atmosphere (Kaufman, 1993). The large offset found in the APEX(438–490 nm) surface reflectances is likely also related to the large deviations from the calibration wavelength of 500 nm.

In Vlemmix et al. (2017), the SBI effective surface reflectance was compared as well with the Landsat 8 surface reflectance product, showing a good agreement for the combination of the morning and afternoon flight data, with a Pearson correlation coefficient of 0.8 and a slope of 1.03. According to this study, considerable differences detected for some of the highest albedo peaks in both data sets might also be related to the fact that exact pixel alignment is crucial and also because bright infrastructural elements may have highly non-uniform bidirectional reflectance distribution functions (BRDFs), which makes the comparison more critical to differences in viewing and illumination angle. Note that the Landsat 8 scene was acquired at 11:56 LT corresponding to an SZA of 42° while for the morning and afternoon flight the SZA varied between 58°–42° and 43°–59°, respectively.

#### 4.2.2.2 AMF dependence on $NO_2$ and aerosol profiles

The authors are aware of the fact that the assumptions made for the well-mixed $NO_2$ and aerosol extinction profile shape, and constant AOT do not take into account the effective variability that can be expected for these constituents in an urban environment. This was already discussed in Vlemmix et al. (2017) for the SBI flights over Berlin and AMF uncertainties related to profile shape and AOT assumptions were estimated to be around 7-10 % based on a set of different scenarios.

In this study, sensitivity tests were performed as well, based on varying $NO_2$ and aerosol extinction profiles, and with the analysis wavelength, surface reflectance, VZA, SZA, and RAA set at respectively 490 nm, 0.05, 7°, 50°, and 90°. Previous studies, such as Leitao et al. (2010) and Meier et al. (2016) indicate that aerosols can enhance or reduce the AMF, depending on their position with respect to the $NO_2$ layer, the optical thickness and the absorption of the aerosol layer. When assuming a well-mixed $NO_2$ and aerosol box profile scenario instead of a Rayleigh atmosphere, AMFs increase by 6 % on average. This can be explained by the urban aerosols with high SSA, which have strongly reflective properties. This causes multiple scattering and an enhancement of the optical path length in the $NO_2$ layer and, thus, results in an increase of the AMF. For the afternoon flight, a scenario was tested with the $NO_2$ layer closer to the sources, extending from the surface to 500 m, and with the aerosols well-mixed in the PBL, extending to 1100 m. In this case, the highly reflective aerosols have a shielding effect as more solar radiation is scattered above the $NO_2$ layer. This results in an overall decrease of 15 % in the AMF when compared to the scenario with both $NO_2$ and aerosols well-mixed in the PBL.

#### 4.2.2.3 AMF dependence on sun and viewing geometries

The dependence of the AMF on sun and viewing geometries is very small under the current conditions and set-up, as can be seen in Fig. 6. Based on a sensitivity study reported in Tack et al. (2017) the strongest effect is expected to originate from the changing SZA, but this is smaller than 6 % for a flight time of 2-3 hours close to local noon in the spring or summer season. The overall AMF at the end of the flight (SZA = 45°) is slightly smaller than at the beginning of the flight (SZA = 60°) due to the smaller SZA, and thus shorter light path through the troposphere. A stronger effect on the AMF is, however, expected in the case of very shallow sun elevation angles.

#### 4.2.2.4 AMF dependence on the sensor altitude

In Fig. 8, the dependence of the AMF on sensor altitude is simulated for five scenarios, based on the concept of box-AMFs. Box-AMFs describe the sensitivity of the observations as a function of altitude, resulting in an assessment of the instrument vertical sensitivity (Wagner et al., 2007). The five scenarios, from low to high altitude, resemble typical platform altitudes of (1) an unmanned aerial vehicle (UAV), (2) the Cessna 207T D-EAFU, (3) the Dornier DO-228 D-CFFU, (4) a potential stratospheric high altitude pseudo-satellite (HAPS) or stratospheric UAV and (5) a sun-synchronous LEO satellite. The sensitivity of the instrument to $NO_2$ is strongly height dependent and is largest for the layer directly under the sensor. Due to scattering and absorption, the sensitivity to $NO_2$ decreases towards the ground surface, where usually most of the

tropospheric $NO_2$ is present due to the proximity to the emission sources. Moreover, the decrease in sensitivity is stronger with increasing platform altitude due to the larger scattering probability above the absorbing layer. The surface box-AMF for the platform altitude of 0.8 km is more than two times larger than the surface box-AMF for a platform altitude of 700 km. Under the assumed RTM parameter settings, the difference in sensitivity to the ground surface is, however, small (< 3 %) between an airborne sensor operating in the stratosphere (HAPS) and a spaceborne sensor. Above airborne platforms, the sensitivity to $NO_2$ is converging rapidly with increasing altitude to a constant box-AMF of 1.6, a value which is close to the geometrical AMF.

Fig. 9 focuses on the box-AMF profiles in the lowest 15 km for the platform altitude of the Cessna and the Dornier, both for a low and high surface reflectance scenario. Besides the platform altitude dependence, also the surface reflectance dependence can be observed. The effect of variability in the surface reflectance is clearly much stronger than variability in the platform altitude.

### 4.2.2.5 AMF dependence on the analysis wavelength

Fig. 10 shows the dependence of the total AMF on the surface reflectance (upper panel) and analysis wavelength λ (lower panel) for both platform altitudes. The AMF dependency to the surface reflectance is clearly non-linear and this is more outspoken for lower albedos. The AMF increases by respectively 65 % and 110 % for the platform altitude at 3.1 and 6.2 km, when increasing the albedo from 1 % to 45 %. Overall, the AMF is larger for the lower platform altitude, however, the AMFs converge to the same value of approximately 3 for very high albedo values.

AMFs also increase with increasing analysis wavelength λ, but the relation seems to be more linear. Note that $NO_2$ is assumed to be optically thin in the visible, not showing any molecular features in the wavelength dependent AMFs. The shorter wavelengths are more affected by Rayleigh scattering than the longer wavelengths, explaining the reduced sensitivity to $NO_2$: photons at shorter wavelengths are scattered more easily before they reach the surface and $NO_2$ layer. The wavelength dependency is slightly stronger for the higher platform altitude. The reduced sensitivity of the APEX instrument, due to the higher platform altitude, is partly compensated by the increased sensitivity due to the fitting interval at larger wavelengths: when considering the same analysis wavelength of 440 nm (middle of the SBI $NO_2$ fitting interval) for the APEX instrument, the sensitivity would increase by 25 % for the altitude at 3.1 km. The increase in sensitivity is only 10 % when considering the analysis wavelength of 490 nm (middle of the APEX $NO_2$ fitting interval) for the APEX instrument.

### 4.3 VCD georeferencing and gridding

Both aircraft are equipped with a navigation system, which records sensor position (i.e. latitude, longitude and elevation) and attitude (i.e. pitch, roll and heading) with high accuracy, allowing for accurate georeferencing of the retrieved VCDs. More details about the navigation system and the georeferencing strategy can be found in Vreys et al. (2016) and Tack et al. (2017) for the Dornier DO-228 D-CFFU and in Meier et al. (2017) for the Cessna 207T D-EAFU. After georeferencing, the $NO_2$ VCDs were gridded in order to generate $NO_2$ distribution maps. For APEX, AirMAP and SBI a regular grid of 0.0011° was

defined, corresponding to a spatial resolution of approximately 120 by 75 m$^2$ along- and across-track. On the other hand, a regular grid of 0.0045° was defined for the SWING retrievals, corresponding to a spatial resolution of 500 by 300 m$^2$. VCDs were assigned based on the pixel center coordinates and multiple VCDs falling into one grid cell were averaged. The chosen grid sizes are slightly larger than the effective spatial resolution of the respective instruments in order to reduce the amount of empty cells in the regular grid. Empty grid cells could occur from sudden changes in roll, pitch and yaw angles during data acquisition. The generated NO$_2$ VCD distribution maps were eventually draped over Google Maps layers in a geographic information system (GIS), QGIS 2.10.1 (QGIS development team, 2009). Note that for the sake of harmonising the different data sets for the quantitative comparison (See Sect. 7), the APEX, AirMAP and SBI retrievals were gridded to the grid size of SWING.

## 4.4 Error budget

The total uncertainty (accuracy and precision) on the vertical column is composed of error sources in (i) the retrieved DSCDs, (ii) the estimation of the residual NO$_2$ amount in the reference spectrum SCD$_{ref}$, and (iii) the computation of the AMFs. Assuming uncorrelated retrieval steps, the contributing error sources are summed in quadrature in order to obtain an estimate of the total NO$_2$ VCD error:

$$\sigma_{VCD_i} = \sqrt{\left(\frac{\sigma_{DSCD_i}}{AMF_i}\right)^2 + \left(\frac{\sigma_{SCD_{ref}}}{AMF_i}\right)^2 + \left(\frac{SCD_i}{AMF_i{}^2} \times \sigma_{AMF_i}\right)^2} \tag{4}$$

We refer to Tack et al. (2017), Meier et al. (2017), Merlaud et al. (2018), and Vlemmix et al. (2017) for in-depth discussions on the retrieval uncertainties of the four respective instruments.

i. The error on the retrieved DSCD or the slant error, $\sigma_{DSCD_i}$, can be estimated from the fit residuals in the DOAS analysis, and is a direct output of it. It is dominated by the shot noise, but it also has a systematic component based on the impact of systematic uncertainties in absorption cross-sections (around 2 % for NO$_2$ (Boersma et al. (2004)) as well as errors due to calibration uncertainties, e.g. slit function and the wavelength calibration. Additional errors result from the use of a NO$_2$ cross-section at a single temperature. As temperatures during the observations were close to the 294 K cross-section temperature, the bias in the tropospheric column is expected to be within 1-2 % (Nowlan et al., 2018). Mean slant errors of 3.3, 2.2, 1.8 and 2.4 × 10$^{15}$ molec cm$^{-2}$ were observed for the APEX, AirMAP, SWING and SBI retrievals, respectively. This is a good approximation for the native slant column detection limit. Note that the whiskbroom SWING instrument has an IFOV of 6°, which is significantly larger that the IFOV of the other instruments. This results on the one hand in an increase of the SNR, when assuming the same effective aperture, as more photons are collected during an observation, but on the other hand in a coarser spatial resolution. This explains the smaller slant column error for SWING

when compared to the other instruments. For the intercomparison study, NO$_2$ VCD maps, retrieved from the different instrumental observations, were all regridded to 0.0045° in order to obtain a similar spatial resolution. This corresponds roughly with the spatial resolution of SWING but is significantly coarser than the resolution of the other instruments. The spatial aggregation results in a decrease of the random uncertainty. Assuming only photon noise, the noise is expected to decrease with the square root of the number of binned data. One SWING pixel corresponds to approximately 17 APEX, 32 AirMAP and 55 SBI pixels which results in a noise reduction by a factor 4, 6 and 7 respectively. Due to the impact of instrumental noise and systematic errors in the DOAS fit, the effective noise is, however, expected to be larger as the noise reduction due to spatial binning is not completely following shot noise statistics. The latter was for example illustrated for the APEX instrument in Tack et al. (2017).

ii. The second error source, $\sigma_{SCD_{ref}}$, originates from the estimation of the NO$_2$ residual amount in the reference spectrum. As no direct measurements at high resolution were performed in the reference area, we assume an uncertainty of 100 % on the estimated NO$_2$ background amount, resulting in a systematic error of $1.0 \times 10^{15}$ molec cm$^{-2}$.

iii. The error on the AMF computation, $\sigma_{AMF_i}$, depends on uncertainties in the assumption of the RTM inputs with respect to the true atmospheric state. The error is treated as systematic (Boersma et al., 2004; Pope et al., 2015; Theys et al., 2017), as it is dominated by systematic errors in the surface albedo, NO$_2$ profile and aerosol parameters. In-depth sensitivity tests were performed in Tack et al. (2017), Meier et al. (2017), Merlaud et al. (2018), and Vlemmix et al. (2017), to study the impact of certain assumptions on the DOAS NO$_2$ retrieval from airborne spectra, such as the assumptions on the surface reflectance, NO$_2$ and aerosol profile. Based on the literature and performed sensitivity tests, discussed in Sect. 4.2.2, the combined uncertainty on the AMF is estimated to be smaller than 20 %.

Mean relative and absolute errors for the retrieved NO$_2$ VCDs are calculated based on the application of the propagation analysis of Eq. (4) on the retrievals, and are provided in Table 5 for the different instruments, for both the morning and afternoon flight. As mentioned earlier, the instrument IFOV can be significantly different and has an impact on the SNR and spatial resolution. For this reason, NO$_2$ VCD errors are provided for both the instrument native resolution and the normalised resolution, used for the intercomparison study. The relative errors are largely in the same range with a minimum of 22 % for SBI and a maximum of 27 % for SWING for the morning flight, and around 23 % for all instruments for the afternoon flight. The absolute errors range from 1.5 to 2.1 x 10$^{15}$ molec cm$^{-2}$ in the morning and from 1.3 to 1.5 x 10$^{15}$ molec cm$^{-2}$ in the afternoon.

Note that a full assessment of the 3D effects of the radiative transport is not done in this study. Taking into account the assumed NO$_2$ layer of 1.1 km (afternoon flight), the relatively large SZAs and the inhomogeneous NO$_2$ field, it is expected that the effective spatial resolution assigned to the VCDs will be reduced by up to 2 orders of magnitude due to 3D

effects of the radiative transport. A full 3D radiative transfer modelling to estimate 1) the effective spatial resolution and 2) errors related to 3D effects of the radiative transport is, however, beyond the scope of this study but will be subject of future work.

## 5 Analysis of the retrieved NO$_2$ VCD map products

The generated NO$_2$ VCD distribution maps are shown in Fig. 11 and 12 for respectively the morning (09:34 - 12:01 LT) and afternoon (14:24 - 16:39 LT) flight on 21 April 2016. Note that all data sets are given the same NO$_2$ VCD color-coding. Note as well that due to practical reasons and time restrictions during the project (time-inefficient retrieval code developed in the framework of a master student graduation project), the first and last two flight lines of the morning flight were not analysed in the processing of SBI level-2 data. The NO$_2$ VCD maps were convolved by a Savitzky–Golay low-pass filter (Savitzky

and Golay, 1964; Schafer, 2011). The filter was only applied for visualisation purposes and, thus, was not used for the quantitative comparison discussed in Sect. 7. Hourly averaged wind profiles were derived with an ADS-B-Receiver, collecting data from ascending and descending aircraft (Bütow, 2016). The Mode-S transponder signals, send out by most airliners, include all necessary information to calculate temperature and wind profiles. The accuracy of the derived profiles was improved by averaging a large number of data points, coming from different aircraft (See Fig. 13). Hourly averaged

wind vectors, indicating the surface wind at flight time, are provided in Fig. 11 and 12. The NO$_2$ horizontal distribution, observed by the different DOAS imagers, is consistent to a high degree. Note, however, the coarser spatial resolution of the SWING grid (see Sect. 3.3 and 4.3) and the non-continuous SBI grid, due to the narrow FOV of the used telescope (see Sect. 3.4).

It is known from emission inventory data (Berlin Senate Department for Urban Development and the Environment,

2017) that an area with strong NO$_x$ emissions is located in the north-western part of the city of Berlin. According to the emission inventory, potential strong NO$_x$ emitters are the power plant "Reuter West" (600 MW) and other industrial facilities close by, as well as the conference center "Messe Berlin". These sites were consequently covered by the flight plan. The wind was blowing from the west and patterns of enhanced NO$_2$ can be clearly observed in the data, which are transported downwind from this area. The NO$_2$ distribution is dominated by an exhaust plume with peak values up to 2 x 10$^{16}$ molec cm$^{-}$

$^2$, crossing the city from west to east, and related to the large power plant "Reuter West". The steam boilers are fired by hard coal and equipped with efficient flue gas scrubbers to generate electricity and heat simultaneously (Vattenfall AB, 2017). The large plume from the power plant is covered for more than 30 km downwind and is continuing towards the east, outside of the acquired region. According to a study of OMI tropospheric NO$_2$ products over the Highveld region in South Africa, such plumes can be sufficiently stable to retain their structure for several hundreds of kilometers downwind (Broccardo et

al., 2018). Enhanced levels of NO$_2$ were indeed observed, approximately 65 km east of Berlin, where the Cessna 207T D-EAFU performed a sounding (not shown).

The plume is clearly confined until it reaches the central part of the city. Then, the plume is broadening towards the east and appears to be more inhomogeneous. This is mostly due to the contribution of emissions from traffic and local sources in the city, but part of the apparent inhomogeneity may be caused by time differences between subsequent flight lines in combination with a dynamically changing $NO_2$ field, as well as the synoptic view of different $NO_2$ layers, which are subject to slightly different wind regimes. As the dominant plume is crossing the city center and ring road, city traffic related $NO_2$ cannot easily be differentiated from it. Examples of differentiating between industrial and traffic emissions have been discussed in earlier studies such as Popp et al. (2012), Meier et al. (2017) and Tack et al. (2017).

Parallel to the Reuter West exhaust plume and just south of it, a second major west-east oriented plume is detected by all DOAS imagers in the morning data. The plume seems to originate from a power and ventilation station at the Messe Berlin conference center. A third clear line source pattern of enhanced $NO_2$ is observed further south-east and seems to be transported from the highways A100 and A113, and industrial buildings surrounding the highways. The $NO_2$ levels are, however, lower than in the two main plumes.

In the southern part of the acquired region, upwind of the city, the pollution levels are much lower due to the lack of major sources in this predominantly sub-urban, rural area. $NO_2$ VCD map statistics are summarised in Table 6: for the morning flight, $NO_2$ levels range between $1 \times 10^{15}$ molec cm$^{-2}$ in the south and $20 \times 10^{15}$ molec cm$^{-2}$ in the dominant plume, with a mean of $7.3 \pm 1.8 \times 10^{15}$ molec cm$^{-2}$. The mean $NO_2$ VCD is relatively low, because of the acquisition of a large background area.

The afternoon data set (see Fig. 12) exhibits largely the same $NO_2$ distribution. Although slightly higher peak values up to $23 \times 10^{15}$ molec cm$^{-2}$ are observed, the mean VCD of $6.0 \pm 1.4 \times 10^{15}$ molec cm$^{-2}$ is lower than for the morning flight. The main exhaust plume, related to the Reuter West power plant, can be observed again. However, the afternoon plume appears to be broken close to the source, which may originate from interruptions in the emissions or plume displacements between overpasses. We checked if two similar looking $NO_2$ hotspots detected in two adjacent flightlines, and indicated by a white asterisk in Fig. 12, could be the same plume feature, transported over the acquisition time of both locations. The measured distance between the two points is approximately 2.3 km. Based on the average wind direction and windspeed of 7 kts and the interval in acquisition time, we determined empirically that the plume feature should have moved over 2.8 km. Differences are expected by variations from the average wind speed and different wind speed at plume height than the assumed surface wind.

The plume is less confined than in the morning and more expanded in north-south direction, which could be related to the weaker wind from the west (around 7 Kts at the surface). The wind direction is also more unstable during the afternoon flight, with the surface wind changing from 301° at 15 LT to 273° at 16 LT and 287° at 17 LT. The slightly different structures observed in the plume, by e.g. APEX and AirMAP could be explained by a combination of (1) the strong spatio-temporal variability of the $NO_2$ field, (2) the delay of up to 20 minutes in acquisition of the $NO_2$ field from the Dornier and the Cessna, and (3) the fact that the maps are built from adjacent flight lines within the time frame of a few hours. Based on the average windspeed of 7 kts and taking into account the delay of up to 20 minutes in acquisition time of the $NO_2$ field, we

estimate that the plume features have been transported over a distance of 4.3 km to the east-southeast within this time interval.

The plume related to the Messe Berlin power station is not detected in the afternoon observations, while the plumes transported from the highways A100 and A113, running south of the city, can be observed again. In the southern part, the background levels seem to increase smoothly to the east. A large artefact is identified in the south (see white dot in Fig. 12), resulting in enhanced APEX $NO_2$ VCDs and decreased AirMAP VCDs. The difference is approximately $1 \times 10^{16}$ molec cm$^{-2}$. The artefact seems to be strongly correlated with a crop field and was identified as winter rape. A possible explanation is that the spectral signature of this crop is spectrally correlated with the $NO_2$ cross-section, affecting the retrievals in a different way depending on the chosen fitting interval. The effect could be similar to the sand/soil signature, discussed in Richter et al. (2011) and Merlaud et al. (2012). Note that a number of smaller similar artefacts are observed in the south, related to the same type of crop.

In general, the $NO_2$ VCD results of the two flights show very similar spatial patterns. All four DOAS imagers allow (1) to retrieve the $NO_2$ horizontal variability at city-scale and (2) to resolve local emission sources. Despite the coarser spatial resolution of SWING, the instrument is able to detect all the relevant patterns of enhanced $NO_2$. The distribution maps show that the $NO_2$ tropospheric columns (1) have an inhomogeneous distribution, (2) can be highly variable and (3) can exhibit strong gradients in an urban context. Due to the relatively coarse spatial resolution of current spaceborne air quality sensors and the local representativeness of ground-based observations, airborne data sets currently provide a unique way to measure and visualise the horizontal distribution of pollutants at the scale of cities.

As mentioned in Sect. 4.4, 3D effects of the radiative transport are not taken into account in this study. It is expected that the effective spatial resolution assigned to the VCDs will be reduced by up to 2 orders of magnitude. Nevertheless, the different data sets will be affected in nearly the same way (same $NO_2$ field, same SZA, but slightly different viewing geometry), reducing the impact of 3D effects of the radiative transport on the intercomparison results of this study.

## 6 Comparison to car-DOAS measurements

The APEX $NO_2$ VCD retrievals of the morning and afternoon flight have been compared with an independent correlative data set acquired by a car mobile-DOAS system, in a similar way as was done for APEX acquisitions over Belgium (Tack et al., 2017), and AirMAP (Meier et al., 2017) and SWING (Merlaud et al., 2018) acquisitions over Romania. During AROMAPEX, mobile car-DOAS measurements were performed by the University of Galati (UGAL), the Max Planck Institute for Chemistry in Mainz (MPIC) and the Royal Belgian Institute for Space Aeronomy (BIRA). In this study, we only validate the APEX $NO_2$ VCDs based on the UGAL car-DOAS observations, as this data set contains most of the $NO_2$ variation, covering background areas as well as large parts of the key $NO_2$ plumes. Note that a harmonization and intercomparison of different car-DOAS observations, performed during several campaigns, including AROMAPEX, is currently ongoing and a full comparison with airborne retrievals will be the focus of a future study.

Details on the instrumental setup of the UGAL zenith-sky car-DOAS system and the $NO_2$ retrieval approach can be found in Constantin et al. (2013). Both in the morning and afternoon, the car followed a route departing from the FUB Institute for Space Sciences building towards the city center and back. The route covered a large part of the major east-west oriented plume. For the comparison, a VCD is extracted from the generated APEX $NO_2$ maps for each co-located mobile measurement. Mobile observations are averaged in case of sampling of the same APEX pixel. The time series of the car-DOAS VCDs are plotted, along with the APEX VCDs at the respective car positions in Figs. 14.a and 14.b for the morning and afternoon flight, respectively.

The time series are in good agreement for both the morning and afternoon, and exhibit largely the same $NO_2$ distribution with low values close to FUB, located in the southwest of the city, and increased levels of $NO_2$ closer to the city center and downwind of the major plumes. Note that gaps in the APEX time series are related to parts of the route outside of the airborne acquisition area. The $NO_2$ VCDs measured along the route during the morning by APEX and car-DOAS are respectively 7.0 and 8.0 x $10^{15}$ molec cm$^{-2}$ on average, and for the afternoon flight 7.3 and 8.5 x $10^{15}$ molec cm$^{-2}$, respectively. The mobile measurements seem to be representative for the whole data set as the averages are close to the mean values for the full $NO_2$ VCD distribution maps, being 7.4 x $10^{15}$ and 6.3 x $10^{15}$ molec cm$^{-2}$ for morning and afternoon, respectively (see Table 6). In general, an overestimation of car-DOAS VCDs or underestimation of APEX VCDs can be observed. This can also be observed in the scatter plots and linear regression analysis, provided in Figs. 15.a and 15.b for the morning and afternoon flight, respectively. The correlation coefficients are 0.86 and 0.96, respectively. For the afternoon flight, the slope and intercept are strongly affected by underestimation of the APEX VCDs between 13:30 and 14:30 LT.

The $NO_2$ column at a certain geolocation is not sampled by both instruments at the same time and variability in local emissions and meteorology can lead to differences. The absolute time offset between car-DOAS and airborne observations can be up to two hours and is provided as well in the scatter plots. There is however not a clear difference in the spread for measurements with a small or large time offset, which lead to the assumption that the $NO_2$ field was relatively stable during the time of measurements.

Efforts were done to ensure the comparability of the correlative data sets, but nevertheless the scatter can be largely explained by sampling of different air masses due to the viewing geometry, differences in the sensitivity to $NO_2$, observation time differences in combination with $NO_2$ variability, and instrumental and algorithmic conceptual differences and related errors and uncertainties. Ongoing work is focusing on harmonization of (1) retrieval settings for the car-DOAS observations and (2) a priori input, e.g. the $NO_2$ profile, aerosols and other properties related to the radiative transfer.

## 7 Intercomparison of the $NO_2$ VCD products

The $NO_2$ VCD maps, retrieved based on data from the different imagers, are quantitatively compared in this section. For the pixel-wise comparison, all $NO_2$ maps were harmonised to ensure comparability and gridded to the same regular grid size of 0.0045°, roughly corresponding to the spatial resolution of the whiskbroom SWING instrument (see Sect. 4.3). Regridding to

a coarser spatial resolution also reduces the impact of fine-scale $NO_2$ differences that can occur due to (1) different sensitivities to $NO_2$, related to the instrumental characteristics, platform altitude, and retrieval algorithm, (2) the slightly different viewing geometries, (3) time differences in the observation of a dynamic $NO_2$ field, and (4) imperfect georeferencing. In order to avoid averaging of measurements from adjacent flight lines, only the central half of the swath has been compared, i.e. for the APEX instrument with a FOV of 28°, only the observations within ± 7° off-nadir have been compared. For APEX, AirMAP and SWING, this corresponds to a swath of roughly 1500 m. These data sets are compared with the full swath of SBI, being 450 m due to its very narrow field of view of 8.3°. The harmonised and intercompared $NO_2$ VCD maps are shown in Fig. 16 for the morning flight.

In Fig. 17, the distribution of the slant errors from APEX and AirMAP retrievals is provided for the morning flight (upper panel). The slant error, $\sigma_{DSCD_i}$, can be estimated from the fit residuals in the DOAS analysis, as indicated in Sect. 4.4. Structures that are correlated with the surface reflectance or the $NO_2$ field cannot be observed. Some flight lines exhibit slightly larger slant errors which is probably related to small instabilities in the spectral performance. For the APEX retrievals, slant errors are generally larger (mean slant error of $3.1 \times 10^{15}$ molec cm$^{-2}$) when compared to AirMAP (mean slant error of $2.1 \times 10^{15}$ molec cm$^{-2}$). The larger slant errors for APEX retrievals, as well as the larger variability, can be attributed to limitations related to the spectral performance of the APEX instrument, i.e. spectral resolution, sampling rate and robustness of the slit function in operational conditions, as discussed extensively in Kuhlmann et al. (2016) and Tack et al. (2017). As most of the fit errors are absolute errors and do not scale with the $NO_2$ signal, the distribution of the relative slant errors (relative to the retrieved slant columns) is provided as well in Fig. 17 (lower panel). The relative slant error is on average 37 % and 24 % for APEX and AirMAP retrievals during the morning flight, respectively. For smaller $NO_2$ abundances, e.g. upwind and south of the city center, the relative error is largest. In the background area, the relative slant error is often very high in case of the APEX observations and retrievals are close to the detection limit. The high retrieval uncertainty in these areas can result in the presence of slightly different structures in the retrieved $NO_2$ VCD maps.

Time series of the pixel-wise VCD comparison are provided in Fig. 18, with APEX data in green, AirMAP in red, SWING in blue and SBI in purple for a) the morning flight and b) the afternoon flight, respectively. All data sets have been compared to the AirMAP $NO_2$ VCDs. Note that the time on the X-axis corresponds with the UTC time recorded by the Cessna 207T D-EAFU, and thus, it is the valid recording time for the AirMAP, SWING and SBI instrument. The absolute time difference between overpasses from the two aircraft was 10 to 12 minutes on average for the morning and afternoon flight, with a maximum difference of 24 minutes. The corresponding scatter plots and orthogonal linear regression analyses are provided in Fig. 19. The color-coding of the lower plots indicates the absolute time offset between the observations from the two aircraft.

The time series exhibit strong $NO_2$ peaks, which correspond to the crossings of the west-east oriented main plume related to the Reuter West power plant. As mentioned in Sect. 2, the flight lines were flown perpendicular to it. The data gaps correspond to the roll movements of the aircraft in order to prepare the acquisition of the next flight line. An overall good agreement can be observed between all observations, both for low and high retrievals. Pearson correlation coefficients

are close to or higher than 0.9 for the morning and afternoon flight, while the linear regression analyses show slopes close to unity and generally small intercepts. As expected, the best agreement is observed between the data sets collected from the same aircraft, as similar air masses were sampled. A very good fit is observed between AirMAP and SBI. This can be partly explained by the fact that the SWING and APEX retrievals contain more noise, mainly due to the instrument characteristics, resulting in a slightly larger spread.

We see a less favorable slope for the VCD comparison between AirMAP and SWING for the afternoon flight. In case of low VCDs, a positive bias can be observed for the first flight lines and an opposite effect for the last flight lines. This is also visible in the $NO_2$ VCD maps (Fig. 12): the west-east oriented smooth increase of the background levels is less present in the SWING retrievals. All SWING retrieval parameters and results were carefully checked and we observed a possible polarisation dependency, which could impact the retrievals. SWING was initially designed to be operated from an UAV, which has repercussions on the size of the instrument. Although a quartz fiber was used, the straight fiber was only 5 cm long which limited its efficiency at depolarising the incident light. Future manned aircraft missions with SWING are planned to be performed with a slightly adapted design, including a longer quartz fiber. As discussed earlier, SWING is a compact instrument without temperature stabilisation or tracking. A temperature dependence could be another possible cause, affecting the retrievals.

## 8 Summary and conclusions

This study presents the first intercomparison of $NO_2$ VCDs, retrieved from four different airborne imaging DOAS instruments. APEX performed flights for the retrieval and high resolution mapping of $NO_2$ VCDs for the first time over Switzerland (Popp et al., 2012) and Belgium (Tack et al., 2017), AirMAP over Germany (Schönhardt et al., 2015) and Romania (Meier et al., 2017), and SWING over Romania (Merlaud et al., 2018). After being tested individually during dedicated campaigns (except for SBI which was deployed here for the first time), the experimental airborne imagers were operated simultaneously over the city of Berlin, in a unique but complex constellation, during the AROMAPEX 2016 campaign. In contrast to APEX and AirMAP, SWING and SBI are compact instruments initially designed to be operated from an UAV and 12-Unit CubeSat, respectively. APEX, AirMAP and SWING have a comparable swath width of 3 km, while SBI has a swath of 450 m. The spatial resolution is better than 100 m for APEX, AirMAP and SBI (pushbroom scanning), and approximately 325 m for SWING (whiskbroom scanning).

The study demonstrates that the $NO_2$ distribution over a large city region can be mapped accurately with high spatial resolution and in a relatively short time frame (typically a few hours). The observations allow to differentiate local emission sources and reveal the fine-scale horizontal variability of tropospheric $NO_2$ in an urban context, eventually contributing to an increased understanding of trace gas distributions and related chemical and dynamical processes in urban areas. For the morning flight (09:34 - 12:01 LT) on 21 April 2016, $NO_2$ levels range between $1 \times 10^{15}$ molec cm$^{-2}$ upwind of the city and $20 \times 10^{15}$ molec cm$^{-2}$ within the dominant plume, with a mean of $7.3 \pm 1.8 \times 10^{15}$ molec cm$^{-2}$. The afternoon data set (14:24 -

16:39 LT) exhibits largely the same horizontal $NO_2$ distribution. Although slightly higher peak values up to 23 x $10^{15}$ molec $cm^{-2}$ are observed, the mean VCD of 6.0 ± 1.4 x $10^{15}$ molec $cm^{-2}$ is lower when compared to the morning flight.

The $NO_2$ VCD products of the four airborne imagers have been qualitatively and quantitatively compared. The data sets are consistent to a high degree after harmonisation of the parameter settings, AMF LUT and gridding algorithm. Pearson correlation coefficients are higher than 0.9, while the linear regression analyses show slopes close to unity and generally small intercepts. This demonstrates the robustness of both the instruments and the applied retrieval approaches. Small discrepancies remain, however, due to a combination of (1) instrumental differences, e.g. SNR, spatial and spectral resolution, and temperature stabilisation, (2) observation differences, e.g. platform altitude, overpass time over a dynamic $NO_2$ field, and viewing geometry, and (3) algorithmic differences, e.g. retrieval of surface reflectance product, and DOAS fitting parameters.

The AROMAPEX study is seen as a preparatory step for forthcoming calibration/validation campaigns for the new generation of spaceborne air quality sensors, such as S-5P, S-4 and S-5. In less than 2.5 h, a (sub-) urban area of approximately 23 by 32 km was covered by the imagers, which is the equivalent of about 30 S-5P/TROPOMI pixels (see Fig. 20). The AROMAPEX study assures a suite of reliable instruments that can be deployed separately from each other for future satellite validation. The high resolution $NO_2$ maps, generated from the airborne data, are unique data sets to study the satellite $NO_2$ intra-pixel variability and to link between global/regional monitoring from space, local air quality models and ground-based observations.

***Acknowledgements.*** The European Space Agency (ESA; contract 4000113511/NL/FF/gp), the European Facility for Airborne Research (EUFAR) and the Belgian Science Policy office (BELSPO; contract BR-121-PI-UAV Reunion) are gratefully acknowledged for funding the AROMAPEX project. The authors wish to express their gratitude to the whole AROMAPEX team and the Freie Universität Berlin (FUB) for their support and cooperation during the campaign.

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

**Table 1.** Flight characteristics of the AROMAPEX data sets, acquired over the city of Berlin.

|  | Morning flight (AM) | Afternoon flight (PM) |
|---|---|---|
| Date (day of year) | 21-04-2016 (112) | 21-04-2016 (112) |
| Flight time LT (UTC + 2) | 09:34–12:01 | 14:24–16:39 |
| # flight lines | 14 | 14 |
| Flight pattern (Heading) | 0°, 180° | 0°, 180° |
| SZA | 58°–42° | 43°–59° |
| Average wind direction | 276° | 285° |
| Average wind speed | 9 Kts | 7 Kts |
| Average temperature | 10° C | 14° C |
| PBL height | 525 m | 1075 m |
| Lat min-max (N) / | 52.35°- 52.55° / | 52.35°- 52.55° / |
| Long min-max (E) | 13.18°- 13.72° | 13.18°- 13.72° |
| Average terrain altitude (a.s.l.) | 70 m | 70 m |

5    **Table 2.** Instrument specifications during the AROMAPEX campaign, defined for APEX for a typical altitude of 6.2 km a.g.l. and for AirMAP, SWING and SBI for a typical altitude of 3.1 km a.g.l. Spatial resolutions are provided after applying spatial aggregation of the APEX and SBI spectra for signal-to-noise enhancement.

|  | APEX | AirMAP | SWING | SBI |
|---|---|---|---|---|
| Wavelength range | 370–2540 nm | 429–492 nm | 280–550 nm | 320–500 nm |
| Spectral resolution (FWHM) | 1.5–3.0 nm | 0.9–1.6 nm | 0.7 nm | 0.3 nm |
| FOV across-track | 28° | 51.7° | 50° | 8.3° |
| IFOV across track | 0.028° | 1.5° | 6° | 0.0051° |
| Swath width | 3100 m | 3000 m | 2900 m | 450 m |
| Ground speed | 72 m s$^{-1}$ | 60 m s$^{-1}$ | 60 m s$^{-1}$ | 60 m s$^{-1}$ |
| Exposure time | 58 ms | 500 ms | 40 ms | 140 ms |
| Across-track spatial resolution | 60 m | 86 m | 325 m | 6 m |
| Along-track spatial resolution | 80 m | 30 m | 325 m | 205 m |
| DSCD detection limit (molec cm$^{-2}$) | ~3.3 x 10$^{15}$ | ~2.2 x 10$^{15}$ | ~1.8 x 10$^{15}$ | ~2.4 x 10$^{15}$ |
| Temperature stabilisation | 19° C | 35° C | No | 25° C |
| Radiometric calibration | Yes | No | No | Yes |
| Weight | 354 kg | 100 kg | 1.2 kg | 8 kg |
| Size (LxWxH) | 83x64x56 cm$^3$ | 92x56x44 cm$^3$ | 33x12x8 cm$^3$ | 31x42x19 cm$^3$ |
| Scanning | Pushbroom | Pushbroom | Whiskbroom | Pushbroom |
| Target platform | Aircraft | Aircraft | UAV | 12-Unit |

**Table 3.** Main DOAS analysis parameters and fitted absorption cross-sections for $NO_2$ DSCD retrieval.

| | APEX | AirMAP | SWING | SBI |
|---|---|---|---|---|
| Wavelength calibration | Solar spectrum (Chance and Kurucz, 2010) | HgCd line lamp/ Solar spectrum (Kurucz et al., 1984) | Solar spectrum (Chance and Kurucz, 2010) | Solar spectrum (Kurucz et al., 1984) |
| Spectral fitting code | QDOAS (Dankaert et al., 2016) | NLIN (Richter, 1997) | QDOAS (Dankaert et al., 2016) | DOAS software TU-Delft |
| Fitting interval | 470–510 nm | 438–490 nm | 425–500 nm | 425–455 nm |
| Cross-sections | | | | |
| $NO_2$ | Vandaele et al. (1998), 294 K | Vandaele et al. (1998), 294 K | Vandaele et al. (1998), 294 K | Vandaele et al. (1998), 294 K |
| $O_3$ | n/a | Serdyuchenko et al. (2014), 223 K | Serdyuchenko et al. (2014), 223 K | Bass and Paur (1985), 225 K |
| $O_4$ | Thalman and Volkamer (2013), 293 K | Thalman and Volkamer (2013), 293 K | Thalman and Volkamer (2013), 293 K | n/a |
| $H_2O$ | n/a | Rothman et al. (2013), 293 K | Rothman et al. (2010), 293 K | n/a |
| Ring effect | Chance and Spurr (1997) | Rozanov et al. (2014) | Chance and Spurr (1997) | Kurucz et al. (1984) |
| Polynomial term | Order 5 | Order 2 | Order 5 | Order 3 |
| Intensity offset | Order 1 | Order 1 | Order 2 | n/a |

**Table 4.** Overview of the input parameters in the radiative transfer model DISORT, characterising the air mass factor look-up table.

| RTM parameter | Grid |
|---|---|
| Wavelength (λ) | 440, 462, 464, 490 nm |
| Sensor altitude (H) | 3080 m, 6230 m a.g.l. |
| Surface reflectance (A) | 0.01–0.35 (steps of 0.01) |
| Viewing zenith angle (VZA) | 0°–30° (steps of 10°) |
| Solar zenith angle (SZA) | 40°–70° (steps of 10°) |
| Relative azimuth angle (RAA) | 0°–180° (steps of 45°) |
| Aerosol optical thickness (AOT) | 0.15 (AM), 0.10 (PM) |
| Aerosol extinction profile (AEP) | $Box_{0.5\,km}$(AM), $Box_{1.1\,km}$(PM) |
| $NO_2$ profile | $Box_{0.5\,km}$(AM), $Box_{1.1\,km}$(PM) |

**Table 5.** Mean NO$_2$ VCD retrieval errors for the morning and afternoon flight. The mean relative errors (percent) and absolute errors (x 10$^{15}$ molec cm$^{-2}$) for the retrieved VCDs are provided for **a)** the native spatial resolution of the different instruments, and **b)** the common resolution of 0.0045° used for the intercomparison study.

| | Morning flight | | Afternoon flight | |
|---|---|---|---|---|
| | a | b | a | b |
| APEX | 36 % (2.7) | 24 % (1.8) | 34 % (2.1) | 24 % (1.5) |
| AirMAP | 29 % (1.9) | 23 % (1.5) | 28 % (1.7) | 23 % (1.4) |
| SWING | 27 % (2.1) | 27 % (2.1) | 23 % (1.3) | 23 % (1.3) |
| SBI | 30 % (2.2) | 22 % (1.6) | 30 % (1.7) | 23 % (1.3) |

10    **Table 6.** NO$_2$ VCD map product statistics for the morning and afternoon flight.

| | Morning flight (x 10$^{15}$ molec cm$^{-2}$) | | | Afternoon flight (x 10$^{15}$ molec cm$^{-2}$) | | |
|---|---|---|---|---|---|---|
| | Mean | Max | St. dev. | Mean | Max | St. Dev. |
| APEX | 7.4 | 19 | 3.6 | 6.3 | 23 | 4.6 |
| AirMAP | 6.6 | 18 | 3.6 | 6.2 | 19 | 4.4 |
| SWING | 7.7 | 21 | 4.1 | 5.7 | 18 | 3.6 |
| SBI | 7.3 | 18 | 3.8 | 5.8 | 20 | 4.2 |

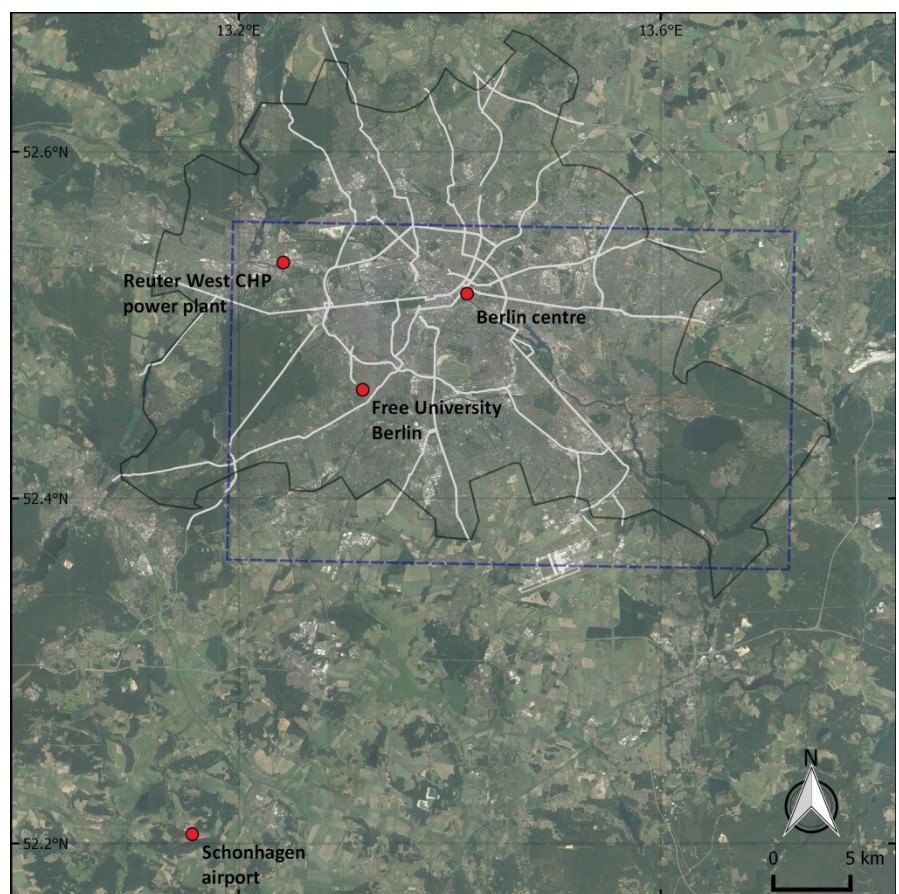

**Figure 1.** Overview map, showing the location of the Berlin city centre, The power plant "Reuter West", the Free University Berlin and the Schönhagen airport. The flight plan is indicated by the blue dashed rectangle. Key roads are shown in white and the city border in black (Google, TerraMetrics).

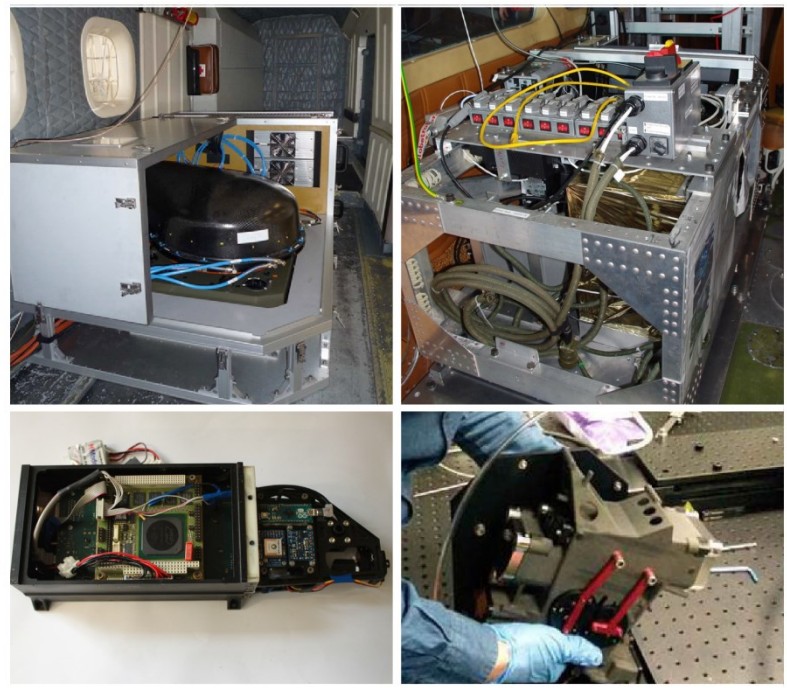

**Figure 2.** Overview of the four DOAS imaging instruments: APEX (top-left), AirMAP (top-right), SWING (bottom-left) and SBI (bottom-right).

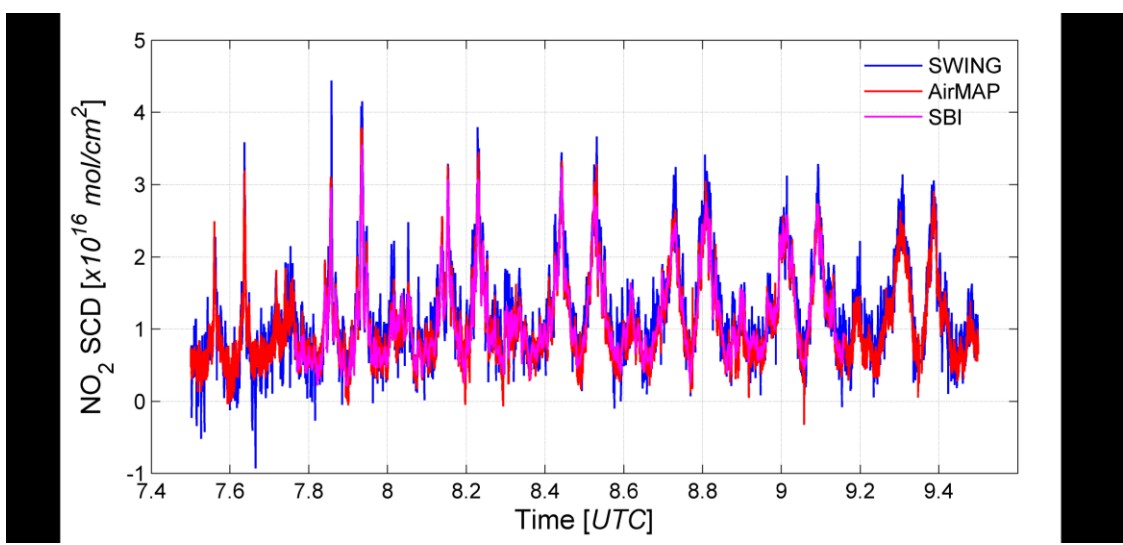

**Figure 3.** Time series of averaged NO$_2$ near-nadir SCDs (> -4° and < 4° VZA), retrieved from AirMAP, SWING and SBI observations, during the morning flight over Berlin on 21 April 2016.

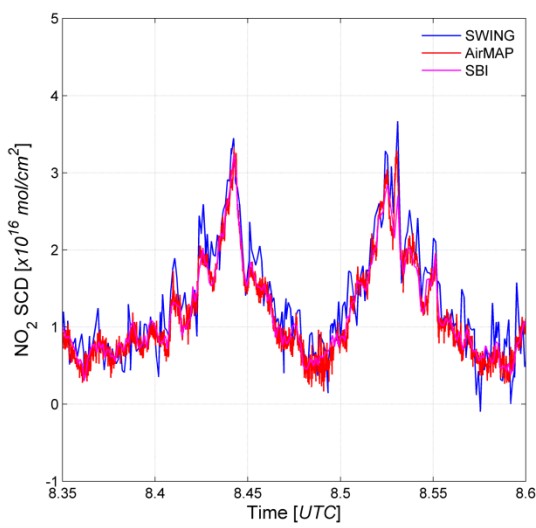

**Figure 4.** Zoom on the NO$_2$ SCD time series of the morning flight over Berlin on 21 April 2016, between 08:21 and 08:36 UTC. The two NO$_2$ peaks correspond to the crossings of the main plume with east-west orientation.

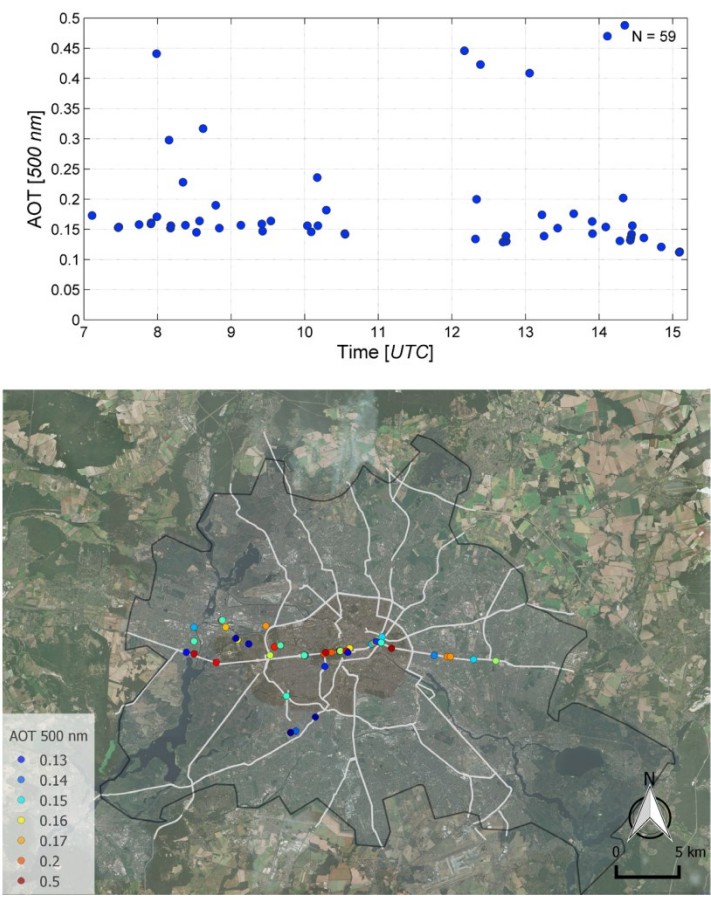

**Figure 5.** Time series (upper panel) and map (lower panel) of AOTs at 500 nm, measured with a Model 540 Microtops II handheld sun-photometer and operated from a car which was driving through the city of Berlin during the aircraft overpasses on 21 April 2016.

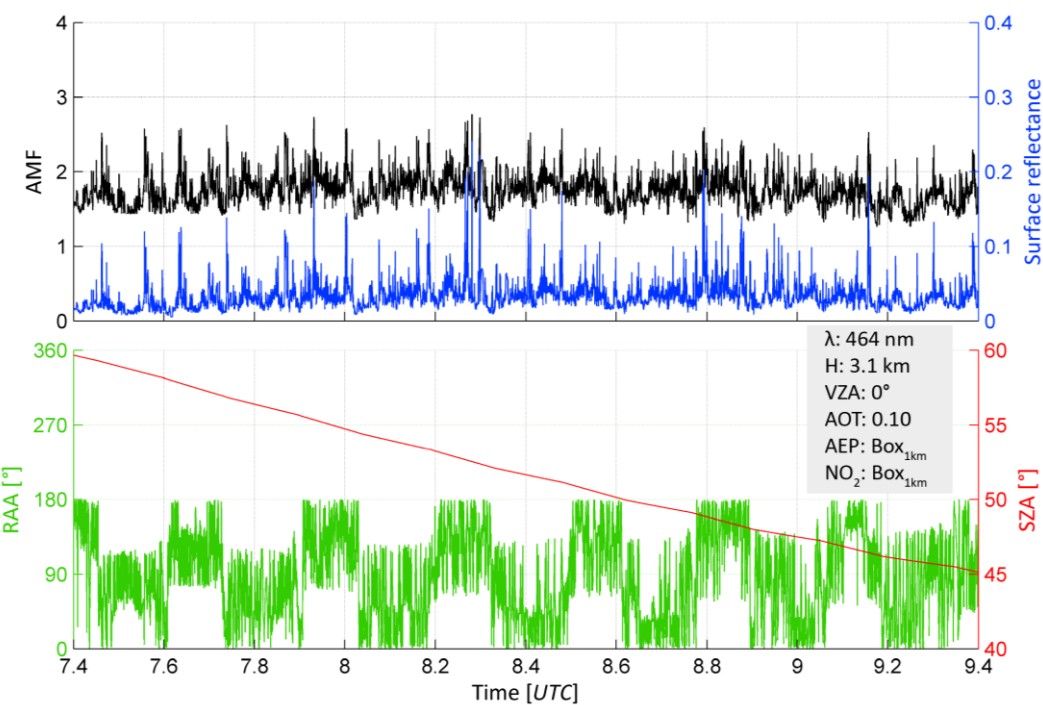

**Figure 6.** Time series of NO$_2$ AMFs for the morning flight on 21 April 2016, computed with DISORT based on the RTM parameters from the AirMAP instrument. The data is plotted for only the nadir observations in each across-track scanline.

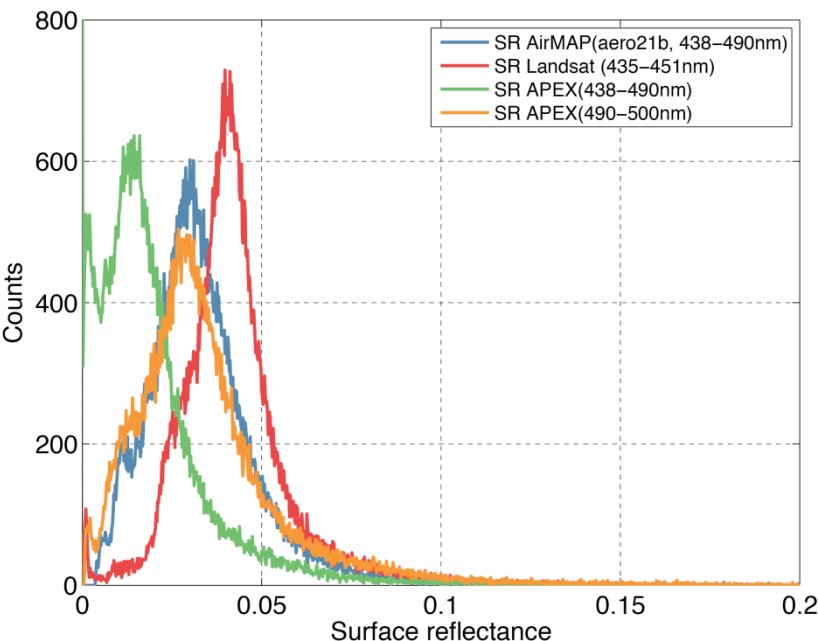

**Figure 7.** Histogram of surface reflectances from AirMAP, APEX and Landsat 8 for the afternoon flight over Berlin on 21 April 2016.

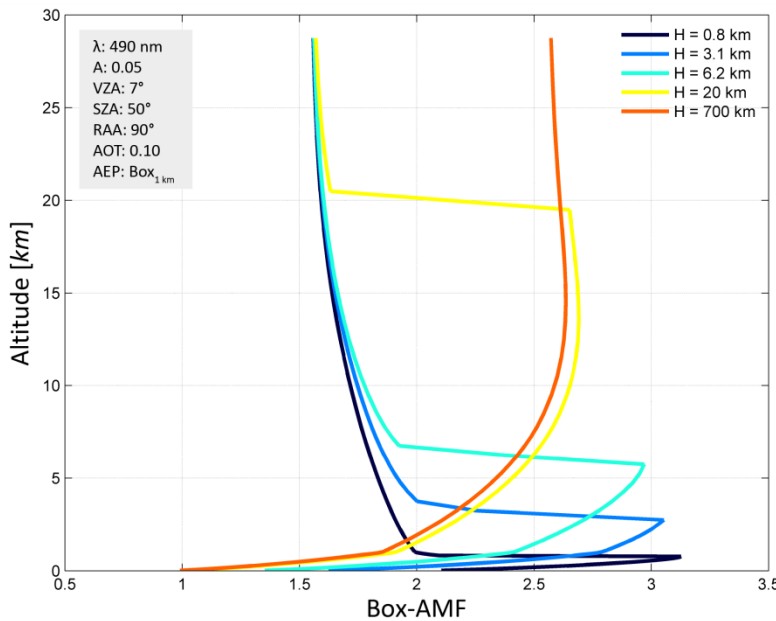

**Figure 8.** Height-dependent box-AMFs assessing the vertical sensitivity to $NO_2$, illustrated for five different scenarios for the sensor altitude H, in km a.g.l.

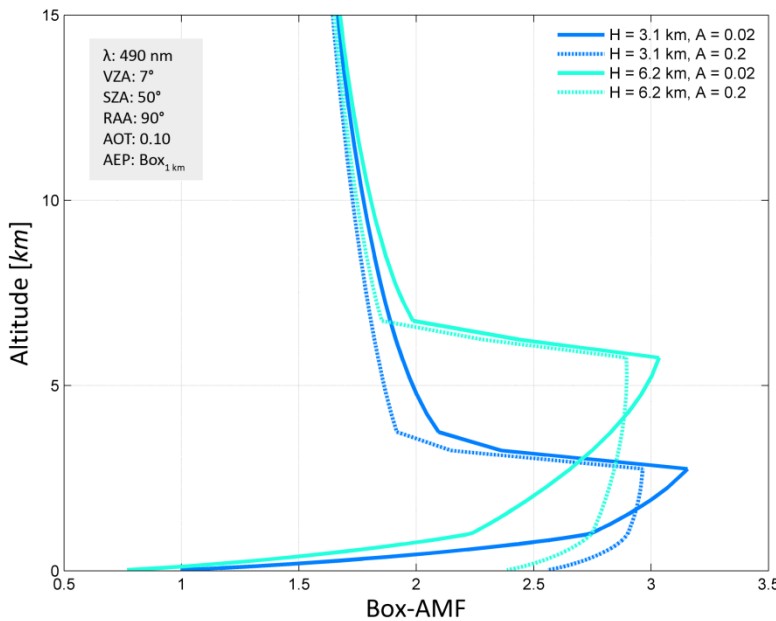

**Figure 9.** Height-dependent box-AMFs assessing the vertical sensitivity to $NO_2$, illustrated for the aircraft altitude of the Cessna 207T D-EAFU (H = 3.1 km a.g.l.) and the Dornier DO-228 D-CFFU (H = 6.2 km a.g.l.), both for a low and high surface reflectance scenario.

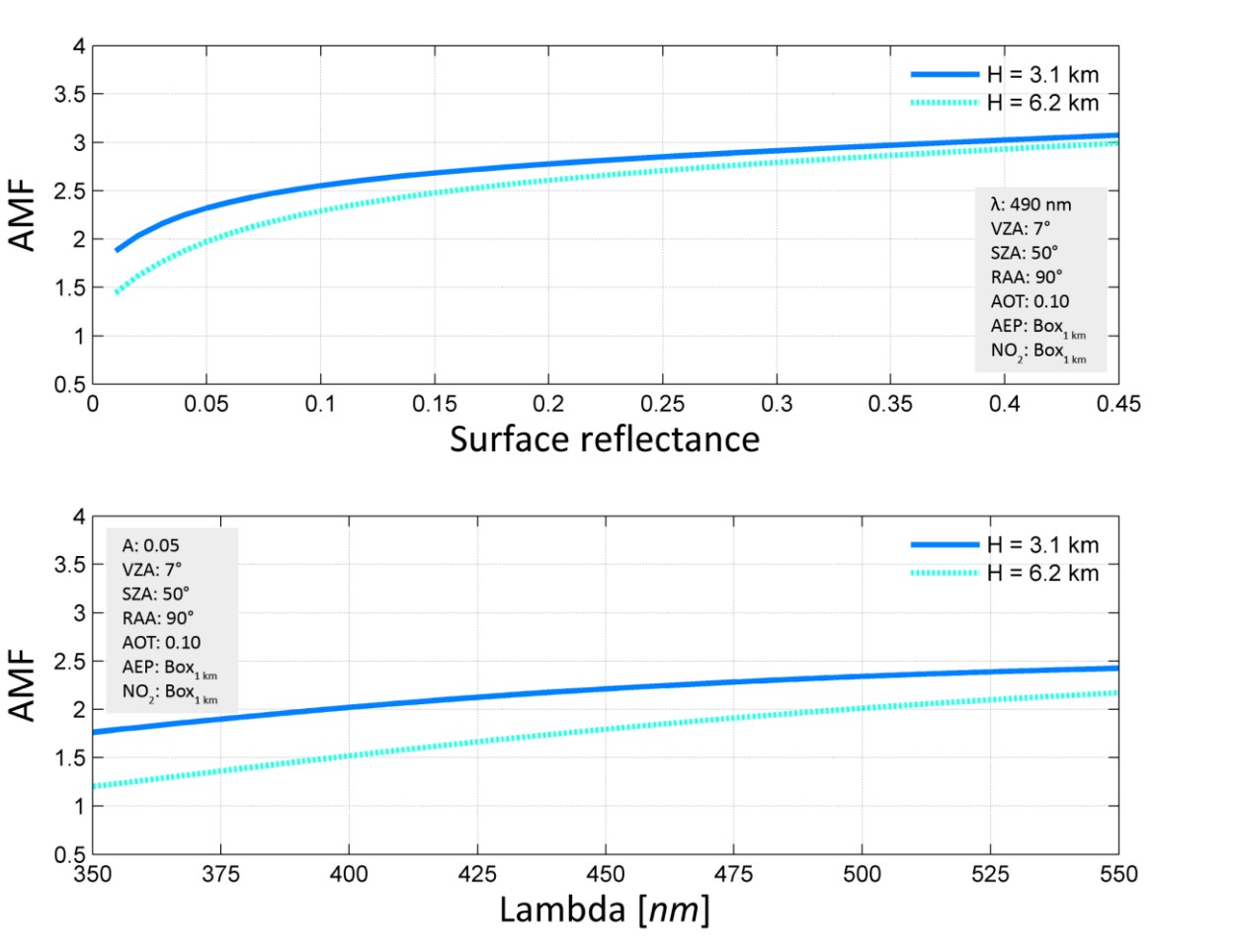

**Figure 10.** Dependence of the AMF on the surface reflectance and RTM computation wavelength (λ) for the aircraft altitude of the Cessna 207T D-EAFU (H = 3.1 km a.g.l.) and the Dornier DO-228 D-CFFU (H = 6.2 km a.g.l.).

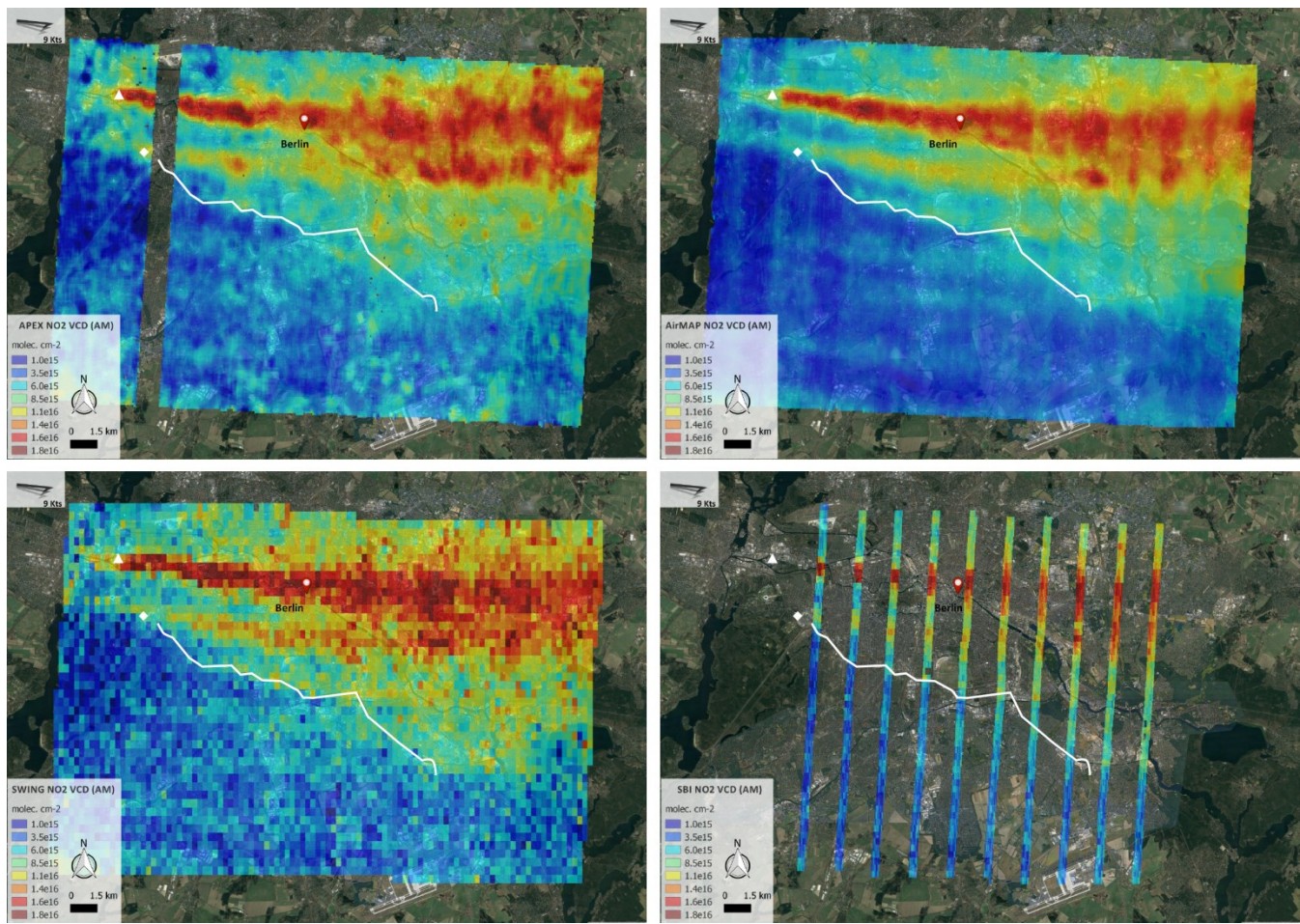

**Figure 11.** Tropospheric NO$_2$ VCD maps retrieved from APEX, AirMAP, SWING, and SBI for the morning flight over Berlin on 21 April 2016 (Google, TerraMetrics). The key contributing NO$_2$ emission sources are indicated by a white triangle (power plant "Reuter West") and white diamond (Messe Berlin). The highways A100 and A113, running south of the city, are marked by the white line. Hourly averaged wind vectors indicate the surface wind at 8:00 (light grey, 6.5 Kts), 9:00 (grey, 9.5 Kts), and 10:00 (black, 10 Kts) UTC. The average surface wind speed is indicated on the maps.

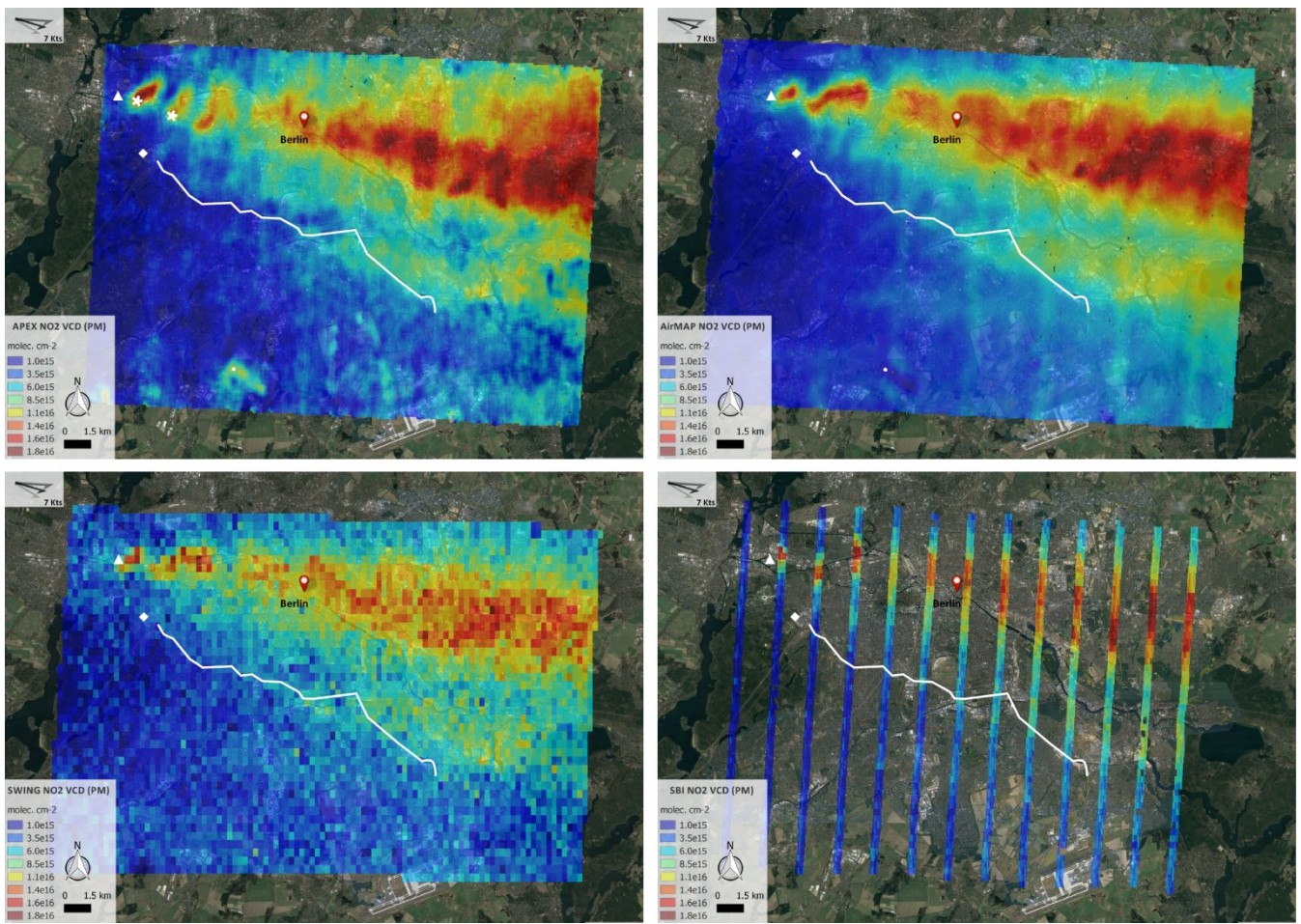

**Figure 12.** Tropospheric NO$_2$ VCD maps retrieved from APEX, AirMAP, SWING, and SBI for the afternoon flight over Berlin on 21 April 2016 (Google, TerraMetrics). The key contributing NO$_2$ emission sources are indicated by a white triangle (power plant "Reuter West") and white diamond (Messe Berlin). The highways A100 and A113, running south of the city, are marked by the white line. Hourly averaged wind vectors indicate the surface wind at 13:00 (light grey, 7.5 Kts), 14:00 (grey, 7 Kts), and 15:00 (black, 7 Kts) UTC. The average surface wind speed is indicated on the maps.

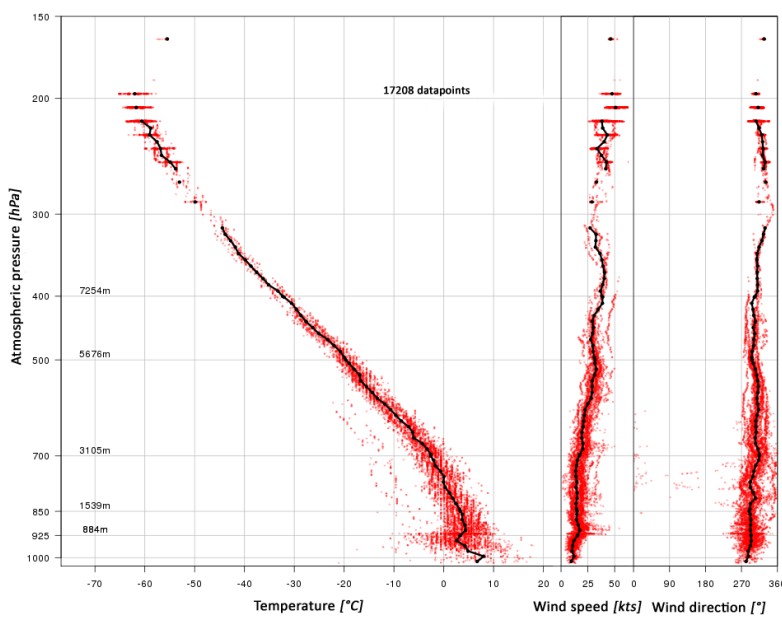

**Figure 13.** Temperature and wind profile on 21 April 2016 at 9 UTC, based on Mode-S-transponder data derived with an ADS-B-Receiver of ascending and descending aircraft in the vicinity of the two Berlin Airports (Bütow, 2016).

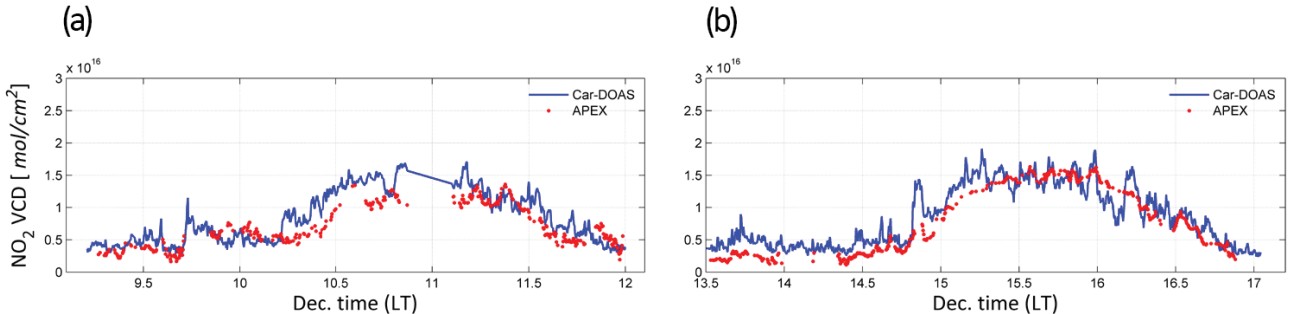

**Figure 14.** APEX and car-DOAS NO$_2$ VCD time series for **(a)** the morning and **(b)** afternoon flight on 21 April 2016, respectively.

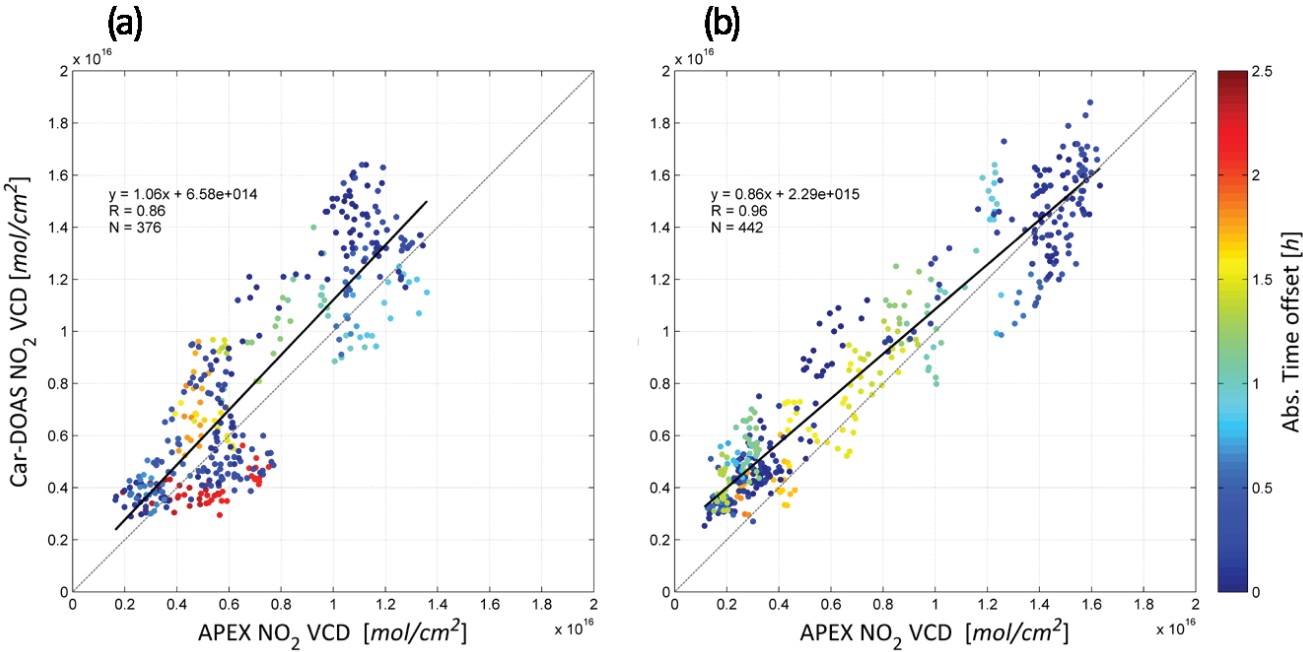

5     **Figure 15.** Scatter plots and linear regression analyses of the co-located NO$_2$ VCDs, retrieved from APEX and car-DOAS for **(a)** the morning and **(b)** afternoon flight on 21 April 2016, respectively. Data points are colour-coded based on the absolute time offset between APEX and car-DOAS observations.

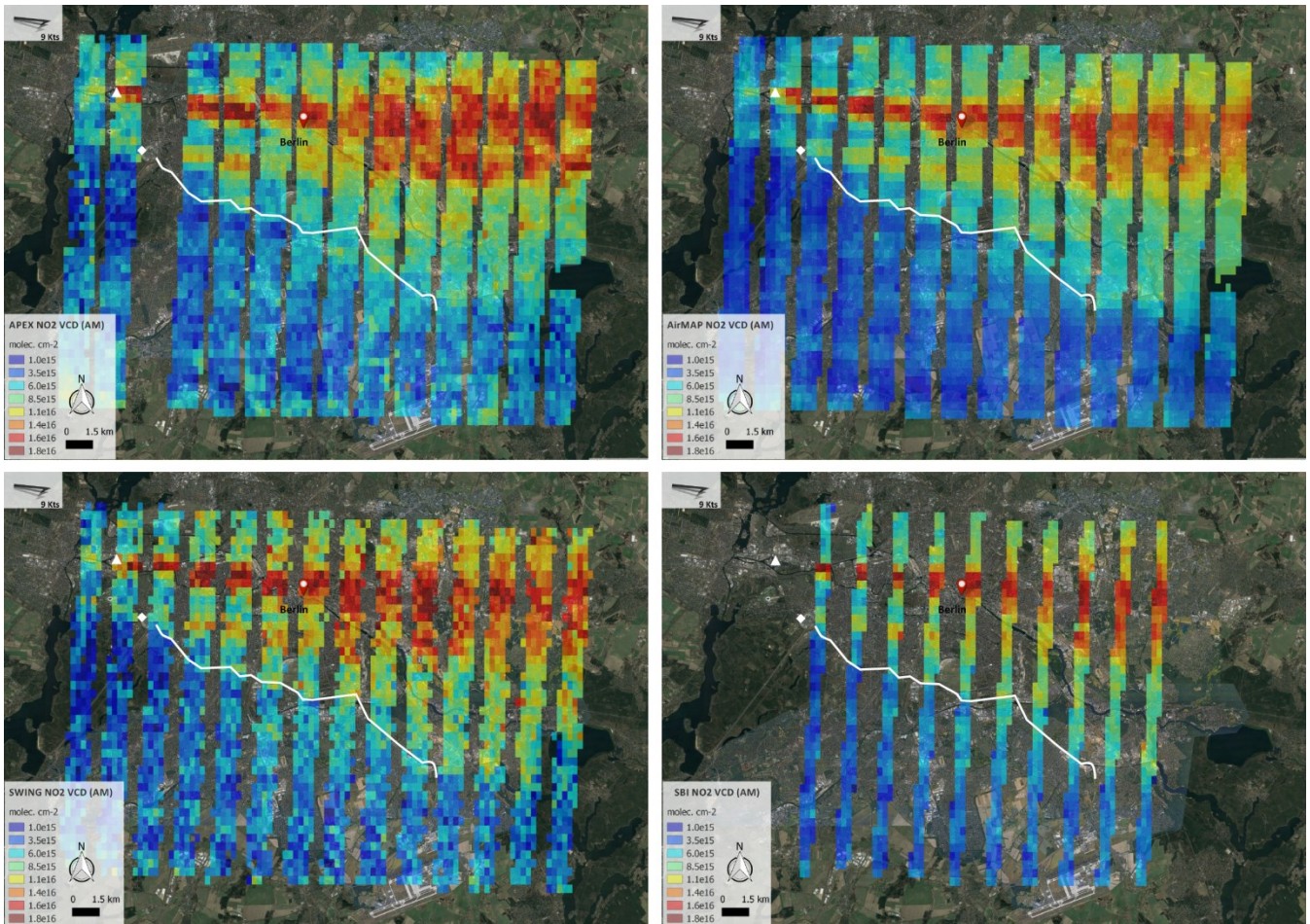

**Figure 16.** Tropospheric $NO_2$ VCD maps retrieved from APEX, AirMAP, SWING, and SBI for the morning flight over Berlin on 21 April 2016 (Google, TerraMetrics). For the pixel-wise comparison, discussed in Sect. 7, all $NO_2$ maps were harmonised to ensure comparability and gridded to the same regular grid size of 0.0045°. Only the central half of the swath has been compared for APEX, AirMAP and SWING, corresponding to a swath of roughly 1500 m. The key contributing $NO_2$ emission sources are indicated by a white triangle (power plant "Reuter West") and white diamond (Messe Berlin). The highways A100 and A113, running south of the city, are marked by the white line. Hourly averaged wind vectors indicate the surface wind at 8:00 (light grey, 6.5 Kts), 9:00 (grey, 9.5 Kts), and 10:00 (black, 10 Kts) UTC. The average surface wind speed is indicated on the maps.

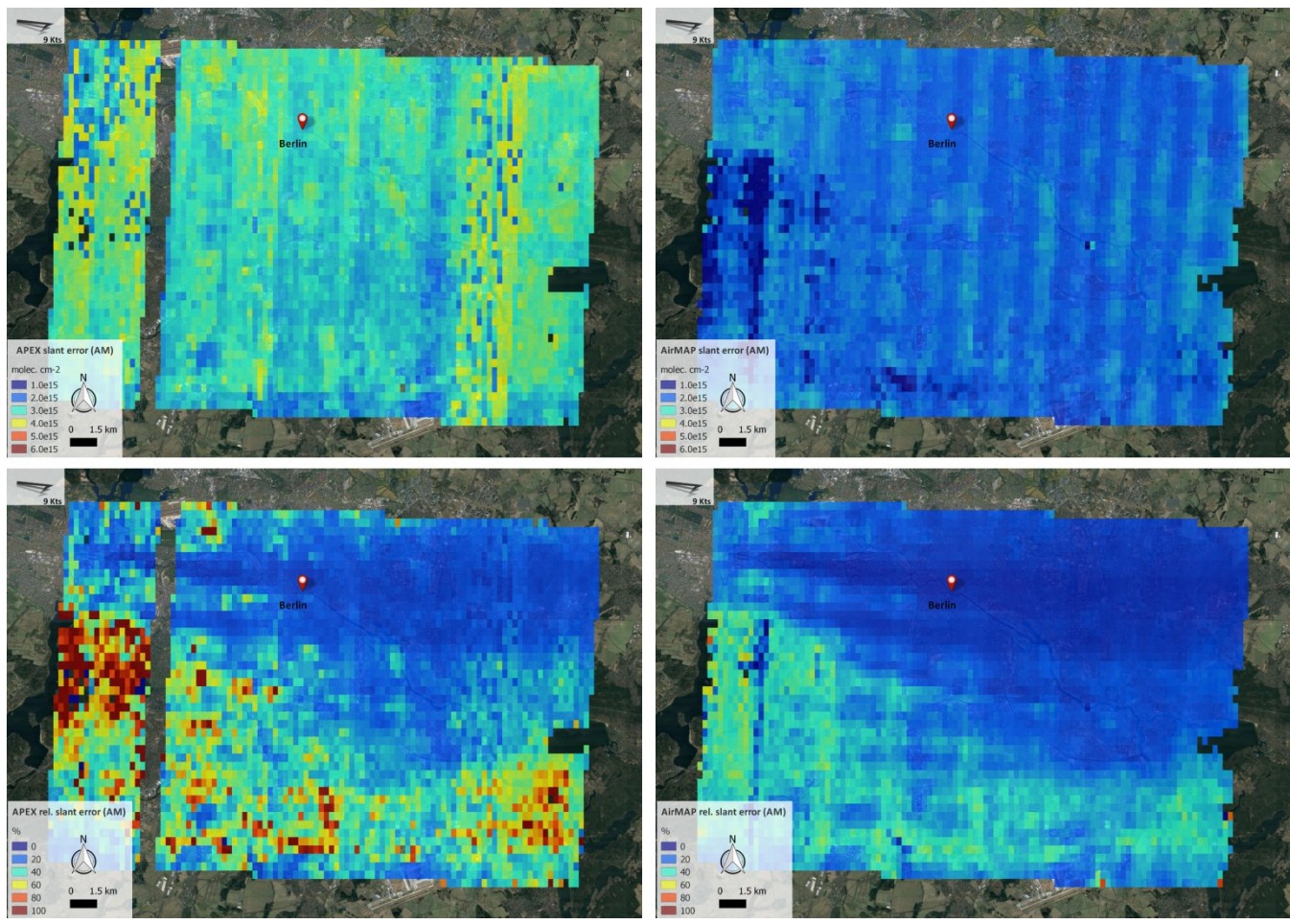

**Figure 17.** Distribution of the errors on the retrieved slant columns from APEX and AirMAP observations (upper panel) and distribution of the relative slant errors for APEX and AirMAP retrievals (lower panel) for the morning flight over Berlin on 21 April 2016 (Google, TerraMetrics).

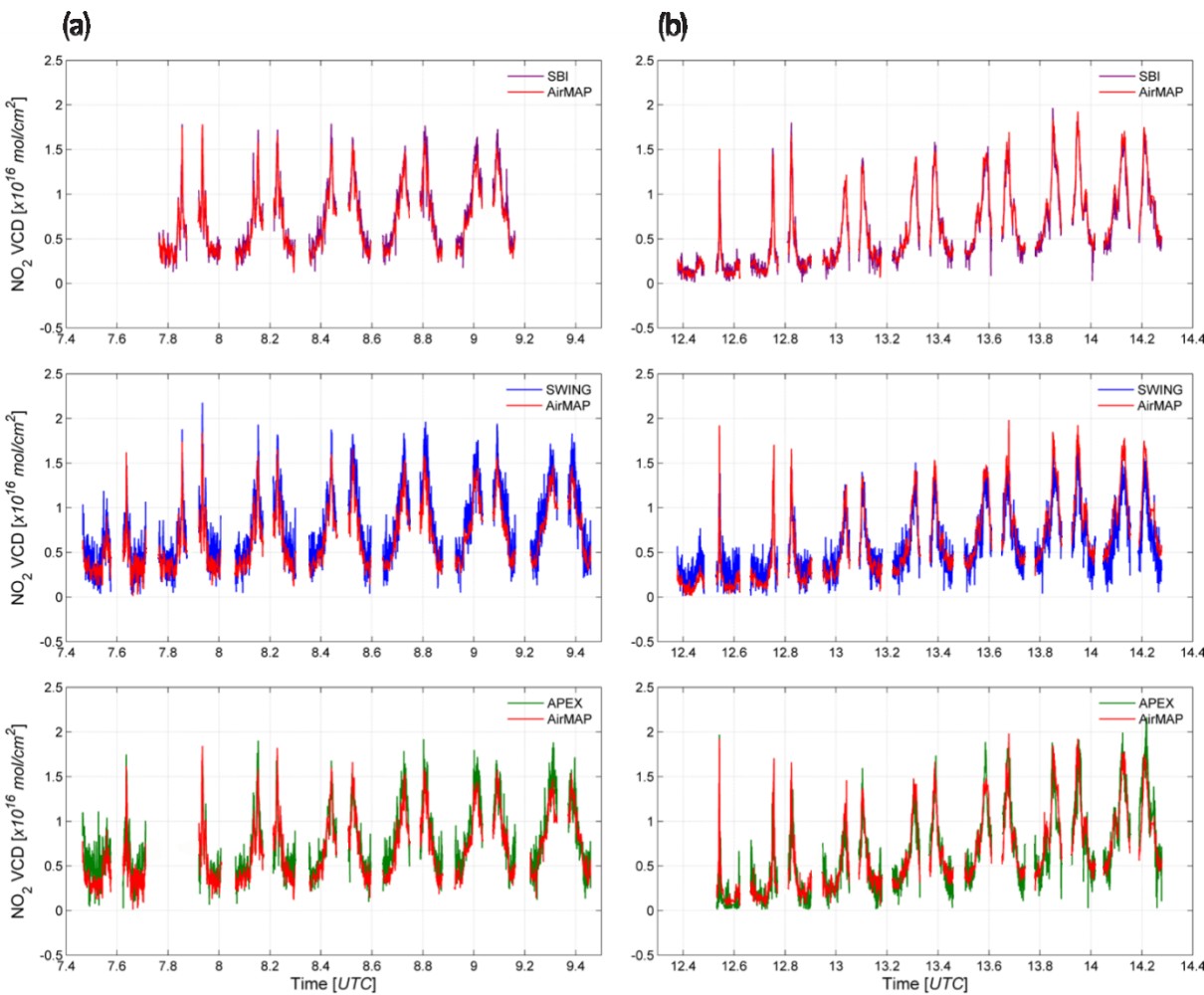

**Figure 18.** Co-located NO$_2$ VCDs retrieved from the harmonised maps for respectively **(a)** the morning and **(b)** afternoon flight on 21 April 2016. APEX VCDs are provided in green, AirMAP in red, SWING in blue and SBI in purple. The X-axis corresponds to the acquisition time, recorded by the Cessna 207T D-EAFU.

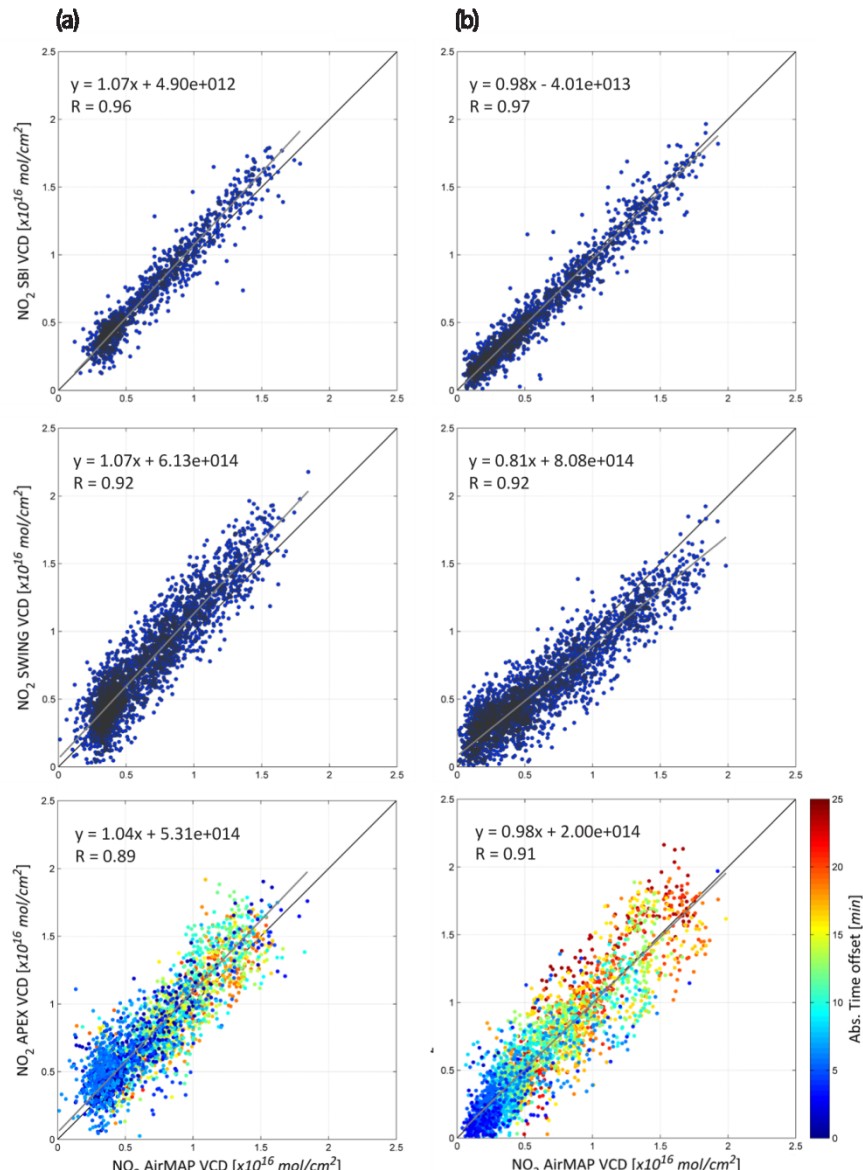

**Figure 19.** Scatter plots and orthogonal linear regression analyses of the co-located NO$_2$ VCDs, retrieved from the harmonised maps for respectively **(a)** the morning and **(b)** afternoon flight on 21 April 2016. The AirMAP NO$_2$ VCDs are plotted on the X-axis. The black solid line and grey line represent the 1:1 ratio and the linear regression, respectively. The colour-coding in the lower plots indicates the absolute time offset between the observations from the two aircraft. Note that the same data points are plotted as in the time series plots of Fig. 18.

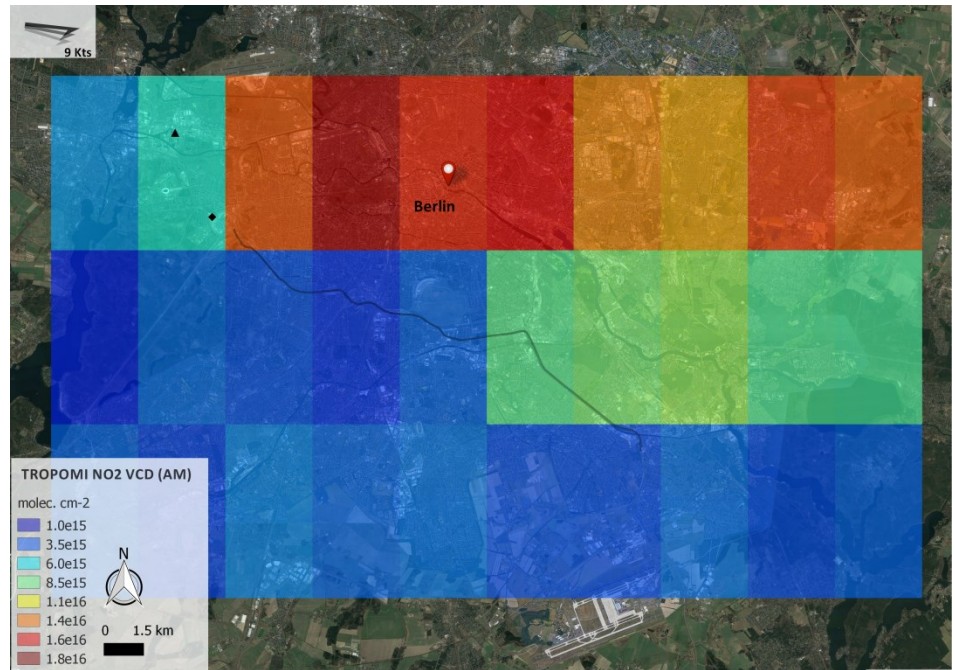

**Figure 20.** Tropospheric NO$_2$ VCDs retrieved from AirMAP for the morning flight over Berlin on 21 April 2016 (Google, TerraMetrics), and gridded at the spatial resolution of the TROPOMI spaceborne instrument (3.5 by 7 km$^2$). The key contributing NO$_2$ emission sources are indicated by a black triangle (power plant "Reuter West") and black diamond (Messe Berlin). The highways A100 and A113, running south of the city, are marked by the grey line. Hourly averaged wind vectors indicate the surface wind at 8:00 (light grey, 6.5 Kts), 9:00 (grey, 9.5 Kts), and 10:00 (black, 10 Kts) UTC. The average surface wind speed is indicated on the maps. A pattern of enhanced NO$_2$ with clear gradients can be observed, originating from the Berlin city region. However, the two main west-east oriented plumes cannot be spatially resolved anymore at the spatial resolution of TROPOMI. Note as well that only a slight NO$_2$ enhancement is observed for the pixel containing the main sources, i.e. the power plant "Reuter West" and Messe Berlin. The plumes are narrow and confined close to the source and the particular pixel contains a considerable amount of background values smoothing out the elevated levels of NO$_2$.