# Peer review of "Intercomparison of four airborne imaging DOAS systems for tropospheric NO2 mapping - The AROMAPEX campaign"

_Atmospheric Measurement Techniques, 2017_

## Referee Comment (RC1) · Anonymous Referee #2 · 4 May 2018

*Comment on 'Intercomparison of four airborne imaging DOAS systems for tropospheric NO2 mapping – The AROMAPEX campaign' by Frederik Tack et al.*

**General Comments**

Tack et al. present the results of the 2016 AROMAPEX campaign. Four different imaging instruments simultaneously recorded reflected/scattered sunlight spectra over Berlin. The retrieved NO2 VCD maps are compared and found to show good agreement.

Similar data of the individual instruments have been published before. However, to the best of my knowledge, the intercomparison of airborne imaging DOAS datasets simultaneously recorded by different instruments hasn't been done before. This might be useful for calibration/validation campaigns for future satellite instruments. The paper also represents a state of the art of airborne NO2 imaging measurements combined in one study. The paper fits the scope of AMT and is well-structured. However, before publication, substantial points need to be clarified and added to the study.

**Specific Comments**

1) VCD spatial resolution

In Sect. 3, the individual imaging instruments are introduced with their respective spatial resolution given by e.g. 80m x 60m for an APEX pixel. It is the surface that is covered by the pixels FOV at ground level. This spatial resolution is then, if I understand correctly, also assigned to the retrieved VCDs. However, in the AMF retrieval 3D effects of the radiative transport are not taken into account. On this high spatial resolution, the assumed NO2 layer of 1km, the large SZAs and the inhomogeneous NO2 distributions, 3D effects will dominate the uncertainties and reduce the effective spatial resolution of the VCD maps (of e.g. APEX and AirMAP) by up to 2 orders of magnitude.

This should be included into the error budget and indicated in the captions of the VCD maps (Fig. 12, 13).

2) Validation with ground based DOAS

The study aims at validating the VCDs of satellite measurements. However, retrievals similar to satellite retrievals are used.

Ground based DOAS measurements can deliver tropospheric VCDs with strongly reduced uncertainty due to their much simpler geometry (e.g. Tack et al., 2015; Brinksma et al., 2008). The reference can be taken at the same location as the airborne reference. The validation of the presented airborne VCDs maps with e.g. zenith mobile DOAS data would drastically increase the scientific quality and significance of the study.

In 'Inter-comparison of airborne atmospheric imagers during the AROMAPEX campaign' (http://www.eufar.net/weblog/2016/06/15/inter-comparison-airborne-atmospheric-imagers-during-aromapex-campaign//, last access: 30.04.2018), Magdalena Ardelean and Alexis Merlaud state that both mobile DOAS and stationary MAX DOAS measurements have been performed during the AROMAPEX campaign.

3) What does the acronym AROMAPEX stand for? If the 'RO' still stands for Romania, why did the campaign take place in Berlin? This might be interesting regarding the submission to the 'AROMAT' special issue.

4) p.2, l.30: Sentinel 5p is already in operation.

5) In Sect. 3, please indicate an approximate detection limit for the dSCDs for the individual instruments in the setup used during the campaign. The VCD maps strongly differ by structures of weaker but still

seemingly significant NO2 VCDs. I think a map with the NO2 fit error or the RMS of the DOAS fit residuals of the individual instruments would be revealing, especially because of the large differences in the DOAS retrieval parameters.

6) Table 3: Why is there no water vapour and O3 absorption cross section used for the DOAS analysis of APEX and no water vapour and no O4 for SBI? Especially when fitting above 500nm the water vapour and O3 absorption cross sections strongly increase. And the SZA differences are significant during a single flight with only one reference.

The results of the DOAS evaluations of e,g, AirMAP could be used to motivate leaving out these species in the evaluations of the other instruments.

7) In Sect 4.1 p.7, l24: 'The differential approach (1) largely reduces systematic instabilities…' compared to what?

8) For the SBI the dataset used in the intercomparison for the morning flight is reduced (only 10 overpasses), while the other instruments deliver data for 14 overpasses. The reason for that should be given.

9) Section 4.2.2.1: The retrieved surface reflectances are compared. AirMAP's surface reflectances are retrieved for the DOAS fit wavelength range and the spatial resolution used in the discussed measurement. They are however compared to two 'APEX surface reflectance products', both having a much higher spatial resolution ('4 by 3 m^2'). As far as I understand, the APEX AMFs are calculated with an 80 by 60 m^2 resolution. Is the high resolution of the surface reflectances taken into account in the retrieval? If not, I would suggest to compare surface reflectances with the spatial resolution of the respective AMF retrieval.

Also the choice of 490-500nm for the surface reflectance retrieval for APEX seems arbitrary and should be motivated (why not 470-510nm?).

10) 4.4 Error budget

a) The argument that a larger FOV per pixel results in more collected photons is only true if all optics use the same effective aperture. The light throughput is determined by the etendue (beam solid angle x effective aperture) of the optics.

b) sigma_scd_ref is included in the error budget as a statistical error. Howerver, it is, as I understand it, an unknown offset. An offset shouldn't be treated as a statistical error.

c) The error analysis should include a discussion of the error introduced by the 3D radiative transfer effects (see Comment 1).

11) p.17, l.12: The artefact in the south of the map is assigned to an eventual spectral structure in the reflection of a specific crop type. This would be interesting. Is there a specific residual structure observed in all affected spectra?

There are significant differences in the DOAS fits used for APEX and AirMAP. Particularly, the APEX fit does not include water vapour, even though the water vapour absorption is much stronger in the APEX evaluation interval compared to the fit interval used for AirMAP. A map of the RMS of the DOAS fit residuals (see Comment 5) would be instructive here.

---

## Referee Comment (RC2) · Anonymous Referee #1 · 7 May 2018

General Comments

This paper presents NO2 retrievals from four airborne remote sensing instruments deployed during the AROMAPEX campaign over Berlin on two flights on 21 April 2016. Details on three of the instruments have appeared in other publications; this was the first deployment of the SBI instrument which is designed to fly on a Cubesat. As all four instruments are likely to become validation instruments for the new generation of air quality satellites (Sentinel 5P, 5, 4), this paper provides a useful intercomparison of analysis approaches, results intercomparisons, error sources, and measurement differences and similarities that will be useful down the road.

[Figure]

The paper is very clearly written, logical and easy to read. I have only a few minor comments to address before I think it can be published in AMT.

Specific Comments

P2, Line 9: Remove brackets around "tropospheric"

P2, Line 10: Remove brackets around "-2" and list GOME and GOME-2 separately. GOME-2 is EUMETSAT also.

P2, Line 11: OMI is Ozone Monitoring Instrument

P2, Line 13: Give rough numbers for resolution for context (ie, tens to hundreds of km)

P2, Line 15: Remove brackets for "higher"

P2, Line 23: "integrate spectroscopy" is a vague expression. Be more specific.

P2, Line 34: Add Zoogman et al., 2017, JQSRT for up-to-date TEMPO reference

P7, Line 14: What temperature cross sections are used? List here or in Table 3. Is there any correction applied to the AMF to account for the temperature dependencies of NO2? Also, this is an additional error term to mention in the error section.

P7, Line 23: Is there a different reference used for each cross track position for each spectrometer? Do you average several spectra together to create each reference? What is effective SNR on the references?What is SZA of reference (close to other measurements?)? What do you use for a VCD_ref value in Equation 3? Is it taken from a model? This also feeds into the error discussion later on. The error says 100% on the reference slant column but I didn't see what that value is or where it came from.

P8, Line 9: I think it would be helpful here to mention fitting noise values (instead of saving only for later error discussion) when discussing different noise levels – helps to interpret the plot.

P8, Line 28: Is the AMF here calculated from the surface to the aircraft or from the

surface to the top of the atmosphere? The high altitude contributions are ignored in Equation 3 but are plotted in later box AMF plots so it's not clear to me.

P 9, Line 23: I think a short description on what route the car drove (or on the map if easy to do) would help the reader interpret this figure. Right now it's not clear where the car was, other than driving through Berlin.

P 10, Line 21: Skipped to Figure 10 from Figure 6, maybe would help to reorder figures to avoid confusion.

P11, Line 18: I have a slightly hard time interpreting this figure or what it means for overall results. I can see you might use it to estimate uncertainty for surface reflectance, but later on you give a flat 20% for AMF uncertainty. You say they agree "well" but not sure what that means (there seems to be quite a large difference in peak location with Landsat, also is 1.47 agreeing well?) What are the sources of near zero values? Are they shadows? Why are there so many APEX values near zero for the shorter wavelengths?

P14, Line 6: Not sure I understand why there are empty grid cells from aircraft attitude changes.

P 14, Line 10 and onwards: Would like to see here and in corresponding table a clear mention of what is systematic/bias causing error and what is random error.

P 15, Line 25: I would like to see original maps, not smoothed just for visualization (could also include that smoothed figure, but smoothing without showing the original data makes me suspicious!)

P 15, Line 30: Skipped discussion back to Fig 11. Maybe move this figure after Fig 12 and 13.

P 17, Line 5: Can you give a number on how far the plume moved based on wind speed and time between flight lines to confirm this could be the source of the difference?

Figure 4: Purple on top of red line is very hard to see.

Figure 10. I am a bit surprised that there are no molecular features visible in the wavelength dependent AMF. (I haven't done the calculation myself – maybe the ozone features would only be visible at shorter wavelengths on this y-axis scale? Just wanted to mention this, to confirm that ozone was properly included in the AMF calculation – if so, ignore this comment.)

Figure 12, 13, 14: Black symbols, grey road are very hard to see. Consider making larger and thicker. Maybe change road color to white if still hard to see. Wind direction is almost impossible to read in plot, but that is okay as is giving in caption.

Technical Corrections

P2, Line 10: SCIAMACHY acronym definition is incorrect

P 12, Line 21: Change "origin" to "originate"

Table 5: Change to "across-track spatial resolution"

---

## Author Comment (AC1) · 2 Nov 2018

**Anonymous Referee #2:**

Thank you for the useful and constructive remarks. As described below, we have modified the manuscript according to suggestions and provided clarifications where necessary. We hope that the revised manuscript has improved in respect to the original paper. Please find a rebuttal against each point below.

*Black, bold, italic: Referee's comments*

Black: Author's reply

Changes in the original discussion paper are highlighted in yellow and attached below

*1) VCD spatial resolution*

*In Sect. 3, the individual imaging instruments are introduced with their respective spatial resolution given by e.g. 80m x 60m for an APEX pixel. It is the surface that is covered by the pixels FOV at ground level. This spatial resolution is then, if I understand correctly, also assigned to the retrieved VCDs. However, in the AMF retrieval 3D effects of the radiative transport are not taken into account. On this high spatial resolution, the assumed NO2 layer of 1km, the large SZAs and the inhomogeneous NO2 distributions, 3D effects will dominate the uncertainties and reduce the effective spatial resolution of the VCD maps (of e.g. APEX and AirMAP) by up to 2 orders of magnitude.*

*This should be included into the error budget and indicated in the captions of the VCD maps (Fig. 12, 13).*

This is a very relevant comment and we are well aware of the 3D effects and the smoothing of the effective spatial resolution. So far a proper study of these effects has not been done in previous publications related to airborne mapping of the atmospheric composition. However, to properly estimate the errors, a full assessment of the 3D effects of the radiative transport, including three-dimensional radiative transfer modelling and $NO_2$ fields from a high resolution 3D CTM are required, which would be beyond the scope of this paper. This is considered to be an interesting but very challenging topic. A future study could focus on a full reanalysis of the AROMAPEX and AROMAT data sets in collaboration with for example LMU (3D RTM MYSTIC) or MPIC (3D RTM McARTIM).

We also assume that as the different data sets will be affected in nearly the same way, the impact of 3D effects of the radiative transport will be small on the intercomparison results of this study.

We should indeed clarify to the reader that 3D effects of the radiative transport are not taken into account in this study.

At the end of Sect. 4.4 we added:

"Note that a full assessment of the 3D effects of the radiative transport is not done in this study. Taking into account the assumed $NO_2$ layer of 1.1 km (afternoon flight), the relatively large SZAs and

the inhomogeneous $NO_2$ field, it is expected that the effective spatial resolution assigned to the VCDs will be reduced by up to 2 orders of magnitude due to 3D effects of the radiative transport. A full 3D radiative transfer modelling to estimate 1) the effective spatial resolution and 2) errors related to 3D effects of the radiative transport is, however, beyond the scope of this study but will be subject of future work."

We also added the following at the end of Sect. 5:

"As mentioned in Sect. 4.4, 3D effects of the radiative transport are not taken into account in this study. It is expected that the effective spatial resolution assigned to the VCDs will be reduced by up to 2 orders of magnitude. Nevertheless, the different data sets will be affected in nearly the same way ( same $NO_2$ field, same SZA, but slightly different viewing geometry), reducing the impact of 3D effects of the radiative transport on the intercomparison results of this study."

*2) Validation with ground based DOAS*

*The study aims at validating the VCDs of satellite measurements. However, retrievals similar to satellite retrievals are used.*

*Ground based DOAS measurements can deliver tropospheric VCDs with strongly reduced uncertainty due to their much simpler geometry (e.g. Tack et al., 2015; Brinksma et al., 2008). The reference can be taken at the same location as the airborne reference. The validation of the presented airborne VCDs maps with e.g. zenith mobile DOAS data would drastically increase the scientific quality and significance of the study.*

*In 'Inter-comparison of airborne atmospheric imagers during the AROMAPEX campaign' (http://www.eufar.net/weblog/2016/06/15/inter-comparison-airborne-atmospheric-imagers-during-aromapex-campaign//, last access: 30.04.2018), Magdalena Ardelean and Alexis Merlaud state that both mobile DOAS and stationary MAX DOAS measurements have been performed during the AROMAPEX campaign.*

Indeed, mobile-DOAS measurements were performed during the acquisition flights by teams of MPIC, UGAL and BIRA. Comparisons with airborne data have been initiated and first results have been presented at a number of meetings/conferences. There are two main reasons why the decision was taken not to include a comparison with mobile data in this particular study. First of all because we wanted to avoid in-depth discussions on the comparison with mobile data, as such comparisons have already been discussed in the related papers, reporting results from the individual involved airborne imagers (see Meier et al., 2017 for AirMAP; Tack et al., 2017 for APEX; Merlaud et al., 2018 for SWING). Secondly, a harmonization and intercomparison of car mobile-DOAS observations, performed during several campaigns (e.g. MADCAT, AROMAT, CINDI-2, AROMAPEX) is currently ongoing and once remaining differences between the operated mobile car-DOAS instruments are well understood, a full comparison with airborne retrievals would be the focus of a future study.

Nevertheless, we agree that adding a comparison with mobile data in this study has added value.

We added a new section 6 to the paper and added also two new figures (Fig. 14 and 15). In our answer below we also provide the APEX $NO_2$ VCD maps for morning and afternoon flight with the

car-DOAS measurements on top of it. To reduce the amount of maps in the manuscript and to avoid adding another data layer to Fig. 11 and 12, we prefer to leave out these maps from the manuscript. All relevant information is present in the time series and scatter plots.

Sect.6:

[revised manuscript text omitted]

**3) What does the acronym AROMAPEX stand for? If the 'RO' still stands for Romania, why did the campaign take place in Berlin? This might be interesting regarding the submission to the 'AROMAT' special issue.**

The acronym "AROMAPEX" is a contraction of AROMAT (Airborne ROmanian Measurements of Aerosols and Trace gases) on the one hand, focusing on operations of the airborne imagers AirMAP and SWING, and APEX on the other hand. However originally not planned, SBI was also added to the set of instruments shortly before the campaign and this was based on own funding. This intercomparison campaign was initially planned to take place in Romania, Bucharest in summer 2015, but was eventually rescheduled to take place over Berlin in spring 2016, due to critical issues with the flight approvals over Romania for the DLR Dornier aircraft. Berlin was an obvious "last minute" choice due to presence of interesting emission sources (already detected during some test flights with AirMAP) and good contacts with air traffic control. However, the consortium stayed largely the same as for AROMAT. AirMAP and SWING activities over Berlin were also funded by the same ESA AROMAT contract.

Clarified at the end of Sect. 2: "The campaign was initially planned to take place in Romania, Bucharest in summer 2015, but was eventually rescheduled to take place over Berlin in spring 2016, due to critical issues with the flight approvals over Romania for the DLR Dornier aircraft."

**4) p.2, l.30: Sentinel 5p is already in operation.**

We suggest to change:

"ESA has planned the Sentinel-5 Precursor (S-5P) and Sentinel-5 (S-5) sun-synchronous low earth orbit (LEO) missions (Ingmann et al., 2012)."

to:

"ESA has launched Sentinel-5 Precursor (S-5P) on 13 October 2017, being a sun-synchronous low earth orbit (LEO) mission (Ingmann et al., 2012), and has planned the launch of the first Sentinel-5 (S-5) in 2021."

*5) In Sect. 3, please indicate an approximate detection limit for the dSCDs for the individual instruments in the setup used during the campaign. The VCD maps strongly differ by structures of weaker but still seemingly significant NO2 VCDs. I think a map with the NO2 fit error or the RMS of the DOAS fit residuals of the individual instruments would be revealing, especially because of the large differences in the DOAS retrieval parameters.*

The APEX, AirMAP, SWING and SBI instruments have a native detection limit of approximately 3.3, 2.2, 1.8 and $2.4 \times 10^{15}$ molec cm$^{-2}$, respectively, with respect to NO$_2$ DSCD retrievals.

A sentence, indicating the detection limit, has been added to each subsection of section 3. The detection limit with respect to DSCD retrievals has also been added as an instrument characteristic to Table 2.

We have also added new plots (Fig. 17) with the distribution of both the absolute and relative slant errors for the APEX and AirMAP retrievals for the morning flight. We have added the following to Sect. 7:

"In Fig. 17, the distribution of the slant errors from APEX and AirMAP retrievals is provided for the morning flight (upper panel). The slant error, $\sigma_{DSCD_i}$, can be estimated from the fit residuals in the DOAS analysis, as indicated in Sect. 4.4. Structures that are correlated with the surface reflectance or the NO$_2$ field cannot be observed. Some flight lines exhibit slightly larger slant errors which is probably related to small instabilities in the spectral performance. For the APEX retrievals, slant errors are generally larger (mean slant error of $3.1 \times 10^{15}$ molec cm$^{-2}$) when compared to AirMAP (mean slant error of $2.1 \times 10^{15}$ molec cm$^{-2}$). The larger slant errors for APEX retrievals, as well as the larger variability, can be attributed to limitations related to the spectral performance of the APEX instrument, i.e. spectral resolution, sampling rate and robustness of the slit function in operational conditions, as discussed extensively in Kuhlmann et al. (2016) and Tack et al. (2017). As most of the fit errors are absolute errors and do not scale with the NO$_2$ signal, the distribution of the relative slant errors (relative to the retrieved slant columns) is provided as well in Fig. 17 (lower panel). The relative slant error is on average 37 % and 24 % for APEX and AirMAP retrievals during the morning flight, respectively. For smaller NO$_2$ abundances, e.g. upwind and south of the city center, the relative error is largest. In the background area, the relative slant error is often very high in case of the APEX observations and retrievals are close to the detection limit. The high retrieval uncertainty in these areas can result in the presence of slightly different structures in the retrieved NO$_2$ VCD maps."

[Figure]

**Figure 17.** Distribution of the errors on the retrieved slant columns from APEX and AirMAP observations (upper panel) and distribution of the relative slant errors for APEX and AirMAP retrievals (lower panel) for the morning flight over Berlin on 21 April 2016 (Google, TerraMetrics).

*6) Table 3: Why is there no water vapour and O3 absorption cross section used for the DOAS analysis of APEX and no water vapour and no O4 for SBI? Especially when fitting above 500nm the water vapour and O3 absorption cross sections strongly increase. And the SZA differences are significant during a single flight with only one reference.*

This has not been discussed for APEX retrievals in this study as this was already treated in Popp et al. (2012) and Tack et al. (2017), discussing more extensively the APEX $NO_2$ retrieval algorithm. Although APEX is initially designed as an airborne hyperspectral imager for land use – land cover (LULC) applications, several studies have demonstrated that the instrument is suitable for atmospheric trace gas retrieval applications, and in particular $NO_2$ (Popp et al., 2012; Kuhlmann et al., 2016; Tack et al., 2017). However, these studies have also revealed some limitations related to the spectral performance, i.e. spectral resolution (2.4-3.3 nm), sampling rate (3.1-3.6 pixels per FWHM) and robustness of the slit function in operational conditions. Interference with unidentified instrumental artefacts or features prevents us from extending the fitting window to wavelengths lower than 470 nm. Currently, the chosen wavelength interval is considered to be the best trade-off between sensitivity to $NO_2$ on the one hand and minimum interference with other absorbers and instrumental structures on the other hand. $O_3$ and $H_2O$ cross-sections were initially fitted in the precursor studies but were eventually left out in order to reduce cross-correlations and overparameterisation of the small fitting interval.

For SBI, on this clear sky day with ground-based AOT measurements available we decided not to attempt to retrieve aerosols, hence to broaden the fit window in order to include $O_4$. Concerning $H_2O$, we saw no indications of patterns in the residuals that correlated with the shape of the water vapour differential cross-section. This is also expected on such a clear sky day over a relatively small region.

We have clarified this in Sect. 4.1:

"Note that $O_3$ and $H_2O$ cross-sections were not fitted in the APEX retrievals due to cross-correlations and overparameterisation of the small fitting interval. $O_4$ and $H_2O$ were not fitted in the SBI retrievals due to small absorption in the chosen fitting window. There were also no patterns visible in the residuals that correlated with the shape of the water vapour differential cross-section. This is also expected on such a clear-sky day over a relatively small region."

**7) In Sect 4.1 p.7, l24: 'The differential approach (1) largely reduces systematic instabilities…' compared to what?**

We cannot take an extraterrestrial reference spectrum $I_0$ with the particular airborne instruments, so based on the DOAS approach we retrieve differential slant columns based on a reference containing a low amount of $NO_2$. This approach reduces systematic instabilities related to instrumental artefacts, and the prominent Fraunhofer lines, present in both spectra.

**8) For the SBI the dataset used in the intercomparison for the morning flight is reduced (only 10 overpasses), while the other instruments deliver data for 14 overpasses. The reason for that should be given.**

SBI performed well during the full length of both the morning and the afternoon flight. The processing of the SBI dataset was done as part of a master student graduation project and the entirely new algorithm that was developed for this purpose was not optimized for processing speed. Towards the end of the project it was decided to reduce the total data volume a bit such that several reprocessings could be done faster. For this reason, the first two and last two tracks/overpasses were not analyzed. A new – more time-efficient – version of the algorithm is currently under development at KNMI, but this is not yet operational.

Added at the beginning of section 5:

"Note as well that due to practical reasons and time restrictions during the project (time-inefficient retrieval code developed in the framework of a master student graduation project), the first and last two flight lines of the morning flight were not analysed in the processing of SBI level-2 data."

**9) Section 4.2.2.1: The retrieved surface reflectances are compared. AirMAP's surface reflectances are retrieved for the DOAS fit wavelength range and the spatial resolution used in the discussed measurement. They are however compared to two 'APEX surface reflectance products', both having a much higher spatial resolution ('4 by 3 m^2'). As far as I understand, the APEX AMFs are calculated with an 80 by 60 m^2 resolution. Is the high resolution of the surface reflectances taken**

*into account in the retrieval? If not, I would suggest to compare surface reflectances with the spatial resolution of the respective AMF retrieval. Also the choice of 490-500nm for the surface reflectance retrieval for APEX seems arbitrary and should be motivated (why not 470-510nm?).*

First of all, a more in-depth comparison (beyond the scope of the paper) of the surface reflectances of AirMAP, APEX and Landsat-8, as well as the methods used can be found in:

Meier, A. C.: Measurements of horizontal trace gas distributions using airborne imaging differential optical absorption spectroscopy, University of Bremen, Bremen, 21 December. [online] Available from: https://elib.suub.uni-bremen.de/peid/D00106465.html, 2017.

In the following, your questions shall be briefly answered. For details please refer to the referenced thesis.

The comparison of surface reflectances between APEX and AirMAP is performed on the native spatial resolution of AirMAP. The high spatial resolution gridded surface reflectance dataset of APEX was sampled using a weighted average of the grid cells covered by an AirMAP footprint (polygon). Using this approach, the compared surface reflectances should show no significant difference caused by different spatial resolutions. Small differences may however be introduced by non-perfectly georeferenced datasets as well as different viewing geometries as directional properties of the surface (BRDF) are not accounted for.

The APEX surface reflectance retrieval uses a calibration wavelength of 500 nm and the retrieved surface reflectances are deviating from the reference measurements with increasing distance from the calibration measurements. This can be seen from the following figure showing the spectral surface reflectance of APEX together with the ground-based reference measurements (ASD FieldSpec) over three calibration surfaces. Please see as well the related reply to question 10 of referee 1.

[Figure]

The choice of the spectral interval of 490-500 nm is based on this figure, as the bias of the APEX surface reflectance is rather small in this wavelength range, but presumably sufficiently large to

average out narrow spectral features that may exist for some surface types. This spectral interval is also close to the middle of the APEX fitting window (470 – 510 nm).

In response to this question and the related question 10 of referee 1 we have clarified some points in Sect. 4.2.2.1 of the manuscript.

**10) 4.4 Error budget**

**a) The argument that a larger FOV per pixel results in more collected photons is only true if all optics use the same effective aperture. The light throughput is determined by the etendue (beam solid angle x effective aperture) of the optics.**

This is correct and clarified in Sect. 4.4.i). However, in Sect. 4.4 we wanted to roughly compare the sensitivity of the different instruments under comparable conditions (similar area acquired in the same time) in an empirical way. This test points out for example that the sensitivity of AirMAP (0.36 × $10^{15}$ molec cm$^{-2}$ after spatial aggregation) is 5 times better than SWING (1.8 × $10^{15}$ molec cm$^{-2}$).

**b) sigma_scd_ref is included in the error budget as a statistical error. However, it is, as I understand it, an unknown offset. An offset shouldn't be treated as a statistical error.**

In principle we should indeed distinguish between random and systematic errors to allow for a more realistic propagation of uncertainties. This is relevant for satellite measurements, since lots of studies are made using averaged data. However, for the data sets of this study, there is no need to do so. The problem with the separation is that in practice it is very difficult (in most cases) to assess whether the uncertainty is random or systematic. Also the other error sources consist of both random and systematic components (See related answer to comment 12 of reviewer 1). This is usually "something in between" and one would have to estimate the covariances matrices (so accounting for spatial correlations on errors) to do it in a proper way.

However, like in most related studies (e.g. Boersma et al., 2004; Pope et al., 2015; Tack et al., 2017; Meier et al., 2017; Theys et al., 2017; Vlemmix et al., 2017; Merlaud et al., 2018), we assume uncorrelated retrieval steps. Summing up all the corresponding error estimates will lead to slightly overestimated error bars or an error budget which is a bit too conservative.

**c) The error analysis should include a discussion of the error introduced by the 3D radiative transfer effects (see Comment 1).**

Please see our reply to Comment 1.

**11) p.17, l.12: The artefact in the south of the map is assigned to an eventual spectral structure in the reflection of a specific crop type. This would be interesting. Is there a specific residual structure observed in all affected spectra?**

**There are significant differences in the DOAS fits used for APEX and AirMAP. Particularly, the APEX fit does not include water vapour, even though the water vapour absorption is much stronger in the APEX evaluation interval compared to the fit interval used for AirMAP. A map of the RMS of the DOAS fit residuals (see Comment 5) would be instructive here.**

No specific residual structure could be observed in the affected spectra. Most likely some specific structures are hiding in the noise but we would need more spectra or average over a longer integration time to be more conclusive on this.

Please so also our reply to comment 5 and 6.

[revised manuscript text omitted]

---

## Author Comment (AC2) · 2 Nov 2018

**Anonymous Referee #1:**

We greatly appreciate the positive feedback from the referee and the constructive comments. As described below, we have modified the manuscript according to suggestions and clarified where necessary. We hope that the revised manuscript has improved in respect to the original paper. Please find a rebuttal against each point below.

***Black, bold, italic: Referee's comments***

Black: Author's reply

Changes in the original discussion paper are highlighted in yellow and attached below

***1) P2, Line 9: Remove brackets around "tropospheric"***

***P2, Line 10: Remove brackets around "-2" and list GOME and GOME-2 separately.***

***GOME-2 is EUMETSAT also.***

***P2, Line 11: OMI is Ozone Monitoring Instrument***

***P2, Line 13: Give rough numbers for resolution for context (ie, tens to hundreds of km)***

***P2, Line 15: Remove brackets for "higher"***

Ok, corrected:

"For about two decades, tropospheric trace gases, such as $NO_2$, have been monitored and mapped at a global scale by spaceborne sensors like ESA's SCIAMACHY (SCanning Imaging Absorption spectroMeter for Atmospheric CHartographY), ESA's GOME (Global Ozone Monitoring Experiment), ESA's/EUMETSAT's GOME-2, and NASA's OMI (Ozone Monitoring Instrument). See for example Richter and Burrows (2002), Beirle et al. (2010), Boersma et al. (2011), Hilboll et al. (2013), Valks et al. (2011) and Bucsela et al. (2013). However, the coarse spatial resolution in the order of a few tens of kilometers of these spaceborne air quality instruments makes them ineffective for studies of the $NO_2$ field at the scale of cities and for resolving individual emission sources."

***2) P2, Line 23: "integrate spectroscopy" is a vague expression. Be more specific.***

We suggest to remove this particular sentence and to change the previous sentence from:

"Here we present the first intercomparison study of NO2 VCDs, retrieved by the differential optical absorption spectroscopy (DOAS) analysis of visible spectra, observed by four different airborne imaging **DOAS instruments**."

to

"Here we present the first intercomparison study of NO2 VCDs, retrieved by the differential optical absorption spectroscopy (DOAS) analysis of visible spectra, observed by four different airborne imaging **spectrometers**."

*3) P2, Line 34: Add Zoogman et al., 2017, JQSRT for up-to-date TEMPO reference.*

OK, added.

*4) P7, Line 14: What temperature cross sections are used? List here or in Table 3. Is there any correction applied to the AMF to account for the temperature dependencies of NO2? Also, this is an additional error term to mention in the error section.*

Temperatures of the used cross-sections are added to Table 3.

A $NO_2$ cross-section at room temperature (294 K) was used for the retrieval of tropospheric VCDs. As effective temperatures during observations were close to this temperature, the bias is expected to be very small and no correction is applied to the AMF. The impact on the stratospheric column is expected to be larger due to the much colder temperature, however, the stratospheric $NO_2$ contribution cancels out in the approach by using a nadir reference spectrum.

The following has been added to the description of the slant error in Sect. 4.4.:

"It is dominated by the shot noise, but it also has a systematic component based on the impact of systematic uncertainties in absorption cross-sections (around 2 % for $NO_2$ (Boersma et al. (2004)) as well as errors due to calibration uncertainties, e.g. slit function and the wavelength calibration. Additional errors result from the use of a $NO_2$ cross-section at a single temperature. As temperatures during the observations were close to the 294 K cross-section temperature, the bias in the tropospheric column is expected to be within 1-2 % (Nowlan et al., 2018). "

Please note that the first sentence has been added in reply to question 12 (see below).

*5) P7, Line 23: Is there a different reference used for each cross track position for each spectrometer? Do you average several spectra together to create each reference? What is effective SNR on the references? What is SZA of reference (close to other measurements?)? What do you use for a VCD_ref value in Equation 3? Is it taken from a model? This also feeds into the error discussion later on. The error says 100% on the reference slant column but I didn't see what that value is or where it came from.*

We agree that we are too short on this topic in the manuscript. We initially preferred, however, to reduce the discussion about the reference spectra in order to avoid a lengthy discussion in this paper, as slightly different strategies were applied to obtain the reference spectrum and to determine the residual amount in the reference spectrum. The respective strategies were already discussed extensively in the related papers, reporting results from the individual involved airborne imagers (see Meier et al. (2017) for AirMAP; Tack et al. (2017) for APEX; Vlemmix et al. (2017), and Merlaud et al. (2018) for SWING). However, we should indeed provide again the references explicitly in this part of the revised manuscript. Despite the fact that we tried to harmonize the acquisitions and retrievals as much as possible, slightly different strategies were applied related to the nature of the different instruments as well as the "standard" methodology adopted by the respective groups,

e.g. for SWING there is only one detector and a reference spectrum was acquired by pointing nadir over a longer integration time and by averaging spectra. For the pushbroom imagers, a reference is required for each detector in order to avoid striping due to 1) optical aberrations and misalignments, and 2) the intrinsic spectral response which is slightly different for each detector. Reference areas were selected at the start of the flight in the west over the forested area and upwind of the city center, where the $NO_2$ amount is low and homogeneously distributed and where the albedo has a low variability. For each flight, new reference spectra were acquired in order to reduce systematic biases. During operation, airborne instruments are typically exposed to changes in environmental conditions (changes in pressure, humidity and temperature, vibrations, mechanical stress, etc.) which can affect the instrument characteristics and degrade its spectral performance. This has been discussed extensively in Kuhlmann et al. (2016) and Tack et al. (2017) for APEX. We also average over several spectra in order to increase the signal to noise, e.g. in case of AirMAP we average 120 spectra (60 sec) reducing the noise to ~$2.0 \times 10^{14}$ molec cm$^{-2}$.

It is assumed that the background spectrum contains a residual $NO_2$ amount of $1 \times 10^{15}$ molec cm$^{-2}$. This value for the background correction is considered to be a typical value for a European summer month as shown in Huijnen et al. (2010). This assumption was also applied in previous studies, e.g. Popp et al. (2012) and Meier et al. (2017). Based on the good agreement between the SCDs (DSCDs + RSCDs) (see Fig. 3 and 4) we assume that only a small error is introduced due to the slightly different applied approaches.

We have clarified this in a concise way in the revised version of the manuscript (see Sect. 4.1) with a clear reference to the respective publications of the individual instruments for more details:

"Reference spectra were acquired over a clean forest area, west (upwind) of the city center, characterized by a low and homogeneous $NO_2$ field and a low albedo variability. In case of a pushbroom imager, a reference spectrum is required for each across-track detector, each having its intrinsic spectral response, in order to avoid across-track biases. For each flight, new reference spectra were acquired in order to reduce systematic biases due to changes in environmental conditions, affecting the instrument characteristics and its spectral performance. Several spectra were averaged in order to increase the SNR of the reference spectrum, e.g. in case of AirMAP 120 spectra were averaged over one minute, reducing the noise to approximately $2.0 \times 10^{14}$ molec cm$^{-2}$. It is assumed that the background spectrum contains a residual $NO_2$ amount of $1 \times 10^{15}$ molec cm$^{-2}$. This value for the background correction is considered to be a typical value for an European summer month as shown in Huijnen et al. (2010). Due to the nature of the different instruments, a slightly different approach was applied for each instrument in order to acquire the reference spectrum. These have been extensively discussed in the related papers, reporting results from the individual involved airborne imagers (see Meier et al. (2017) for AirMAP, Tack et al. (2017) for APEX, Vlemmix et al. (2017) for SBI, and Merlaud et al. (2018) for SWING)."

***6) P8, Line 9: I think it would be helpful here to mention fitting noise values (instead of saving only for later error discussion) when discussing different noise levels – helps to interpret the plot.***

Approximate detection limits for the different instruments are provided now for each individual instrument in each subsection of sect. 3 based on the comment 11 of the second referee. The detection limit with respect to DSCD retrievals has also been added to Table 2. A sentence has also been added to refer to Sect. 4.4, where the dependence of the effective noise on the averaging of observations is discussed more in detail.

***7) P8, Line 28: Is the AMF here calculated from the surface to the aircraft or from the surface to the top of the atmosphere? The high altitude contributions are ignored in Equation 3 but are plotted in later box AMF plots so it's not clear to me.***

Box-AMFs were calculated from the ground surface till TOA and also used for the sensitivity studies. However, total AMFs, used to calculate the $NO_2$ VCDs, only consider the box-AMFS until approximately the aircraft altitude, or more precisely until top of the assumed $NO_2$ and aerosol extinction box profile, as described in table 4 for AM and PM flight. The stratospheric contribution (and part of the free troposphere) is indeed assumed to be cancelled out because of the small time difference between the reference spectrum and the analysed spectra. The diurnal increase of the $NO_2$ stratospheric column between 80° SZA sunrise and sunset is estimated to be approximately 1 x $10^{14}$ molec cm-2 per hour (Tack et al., 2015 and Tack et al., 2017)

***8) P 9, Line 23: I think a short description on what route the car drove (or on the map if easy to do) would help the reader interpret this figure. Right now it's not clear where the car was, other than driving through Berlin.***

The following map has been added to clarify where the measurements were acquired.

[Figure]

We have clarified this in the manuscript: "In Fig. 5, a time series of retrieved AOTs at 500 nm is shown in the upper panel and a map is provided in the lower panel. Two similar routes were followed in the morning and afternoon, starting from the FUB Institute for Space Sciences."

***9) P 10, Line 21: Skipped to Figure 10 from Figure 6, maybe would help to reorder figures to avoid confusion.***

We prefer to keep the upper panel of Fig. 10 together with the lower panel and have its main discussion in 4.2.2.5, while as well referring to the upper panel in 4.2.2.1. For clarity we refer to "upper panel " and "lower panel" in the text for Fig. 10.

**10) P11, Line 18: I have a slightly hard time interpreting this figure or what it means for overall results. I can see you might use it to estimate uncertainty for surface reflectance, but later on you give a flat 20% for AMF uncertainty. You say they agree "well" but not sure what that means (there seems to be quite a large difference in peak location with Landsat, also is 1.47 agreeing well?) What are the sources of near zero values? Are they shadows? Why are there so many APEX values near zero for the shorter wavelengths?**

First of all, a more in-depth comparison of the surface reflectances of AirMAP, APEX and Landsat-8, as well as the methods used can be found in:

Meier, A. C.: Measurements of horizontal trace gas distributions using airborne imaging differential optical absorption spectroscopy, University of Bremen, Bremen, 21 December. [online] Available from: https://elib.suub.uni-bremen.de/peid/D00106465.html, 2017.

As such an extensive study is beyond the scope of the paper, all details are not provided in the manuscript. However, a reference to the now published thesis is now provided in the paper manuscript.

In the following your questions shall be briefly answered. For further details please refer to the referenced thesis:

The near zero values are likely related to the assumptions made on the parameters in the atmospheric correction. For example, if the scattering by aerosols is assumed larger than it actually is, this will result in a low biased surface reflectance, because the contribution of the atmosphere to the recorded radiance is overestimated. This effect is most pronounced above dark areas, when the relative contribution of light reflected at the surface to the radiance recorded at the aircraft is low. On a map, these low surface reflectance values occur at the lake in the East and to some extent in the forest in the West.

The parameters for the atmospheric correction of APEX surface reflectances were tuned to match ground-based reference measurements performed with an ASD field spectrometer above three different surface types at calibration wavelength of 500 nm (close to the middle of the APEX fitting window). The figure below shows the spectral surface reflectance of these measurements and indicates low biased APEX surface reflectances at wavelengths smaller than the calibration wavelength. This also explains the large amount of near zero values for the APEX retrievals in the 438-490 nm window.

[Figure]

For this reason, the APEX surface reflectance in the AirMAP fit window (438-490 nm) is not considered as a reliable dataset. In order to have a comparison dataset, a spectral interval close to the calibration wavelength (490-500 nm) was chosen. Although the spectral surface reflectance may show small variations between these two spectral regions, it gives reasonable results, when considering the mean of the observed surface reflectances. We agree, that a fitted slope of 1.47 does not seem to "agree well" when this value is considered as isolated metric . However, as stated in the manuscript, the large slope likely originates from the lower dynamic range of the AirMAP values, which results in a strong impact of the smallest and highest values on the fitted slope parameter. The histogram (Fig. 7) shows that for the vast majority of points there is a good agreement between AirMAP / APEX ( 490-500nm).

The Landsat-8 surface reflectance is offset from APEX/AirMAP by about 0.01. If surface reflectance values were sampled from Landsat-8 scenes, the larger surface reflectance values would result in larger AMFs and consequently in smaller $NO_2$ VCDs. For a typical observation scenario and the offset of 0.01, this effect is in the order of 10%.

All key error sources on the AMF were summed in quadrature, providing us an overall AMF uncertainty which is indeed smaller than 20 %. This is also based on the sensitivity tests performed by the co-authors in previous studies to which we refer in Sect. 4.4.

In response to this question and question 9 of referee 2 we have clarified some points in Sect. 4.2.2.1 of the manuscript.

***11) P14, Line 6: Not sure I understand why there are empty grid cells from aircraft attitude changes.***

Due to turbulence, sudden attitude changes (roll, pitch and yaw angles) can occur during data acquisition. As mentioned on p. 14, Line 3, for the mapping of the data, retrieved VCDs were assigned to a cell of the regular grid, based on the pixel center and multiple VCDs falling into one grid cell were averaged. In a perfect world there should be one VCD retrieval per regular grid cell, however, it

occurs that no pixel centers are falling into a certain cell of the regular grid, resulting in a nodata value.

We suggest to change:

"The chosen grid sizes are slightly larger than the effective spatial resolution of the respective instruments. This choice was made in order to reduce the number of empty grid cells due to aircraft attitude changes."

to:

"The chosen grid sizes are slightly larger than the effective spatial resolution of the respective instruments in order to reduce the amount of empty cells in the regular grid. Empty grid cells could occur from sudden changes in roll, pitch and yaw angles during data acquisition."

**12) P 14, Line 10 and onwards: Would like to see here and in corresponding table a clear mention of what is systematic/bias causing error and what is random error.**

We have specified the type of the different error components in Sect. 4.4.:

…"The error on the retrieved DSCD or the slant error, $\sigma_{DSCD_t}$, can be estimated from the fit residuals in the DOAS analysis, and is a direct output of it. It is dominated by the shot noise, but it also has a systematic component based on the impact of systematic uncertainties in absorption cross-sections (around 2 % for $NO_2$ (Boersma et al. (2004)) as well as errors due to calibration uncertainties, e.g. slit function and the wavelength calibration. Additional errors result from the use of a $NO_2$ cross-section at a single temperature. As temperatures during the observations were close to the 294 K cross-section temperature, the bias in the tropospheric column is expected to be within 1-2 % (Nowlan et al., 2018)."….

…"The second error source, $\sigma_{SCD_{ref}}$, originates from the estimation of the NO2 residual amount in the reference spectrum. As no direct measurements at high resolution were performed in the reference area, we assume an uncertainty of 100 % on the estimated NO2 background amount, resulting in a systematic error of 1.0 × 1015 molec cm−2"….

For the AMF error, it is not straightforward to specify it as a strictly systematic or random error. After discussing with different colleagues it is clear that this is subject to heavy debate. However, based on the literature we suggest to treat the AMF error as systematic: the AMF uncertainty will be dominated by systematic errors in the surface albedo, $NO_2$ profile and aerosol parameters. Also, the literature does not provide an estimate of the random error contribution to the AMF uncertainty.

…"The error on the AMF computation is treated as systematic (Boersma et al., 2004; Pope et al., 2015; Theys et al., 2017), as it is dominated by systematic errors in the surface albedo, $NO_2$ profile and aerosol parameters"…

As Table 5 is providing "total" relative and absolute errors for the retrieved $NO_2$ VCDs, and not the different components, we did not specified the error type again here.

Please see as well the discussion related to comment 10.b of reviewer 2.

**13) P 15, Line 25: I would like to see original maps, not smoothed just for visualization (could also include that smoothed figure, but smoothing without showing the original data makes me suspicious!)**

This is the main reason why we provided the maps in Fig. 14. These are the data sets which were effectively compared in the quantitative analysis, and as mentioned in the paper, not convolved by the S-G filter.

**14) P 15, Line 30: Skipped discussion back to Fig 11. Maybe move this figure after Fig 12 and 13.**

Indeed Fig. 11 need to be after Fig. 12 and 13. This is corrected in the revised version.

**15) P 17, Line 5: Can you give a number on how far the plume moved based on wind speed and time between flight lines to confirm this could be the source of the difference?**

In the afternoon, the average surface windspeed was 7 kts or 3.6 m/s. Taking into account the delay of up to 20 minutes in acquisition of the $NO_2$ field from the Dornier and the Cessna, the plume is expected to be transported over a distance of 4.3 km to the east-southeast within this time interval.

We also checked if two similar looking $NO_2$ hotspots detected in two adjacent flightlines (and indicated by a white asterisk in Fig. 13) could be the same plume feature, transported between the acquisition time of both locations. The measured distance between the two points is ~2.3 km. Based on the average wind direction and windspeed of 7 kts and the interval in acquisition time, we determined empirically that the plume should have moved over 2.8 km. Differences can be explained by variations from the average wind speed and different wind speed at plume height than the assumed surface wind.

We have added the following to Sect. 5:

"We checked if two similar looking $NO_2$ hotspots detected in two adjacent flightlines, and indicated by a white asterisk in Fig. 12, could be the same plume feature, transported over the acquisition time of both locations. The measured distance between the two points is approximately 2.3 km. Based on the average wind direction and windspeed of 7 kts and the interval in acquisition time, we determined empirically that the plume feature should have moved over 2.8 km. Differences are expected by variations from the average wind speed and different wind speed at plume height than the assumed surface wind."
As well as:

"Based on the average windspeed of 7 kts and taking into account the delay of up to 20 minutes in acquisition time of the $NO_2$ field, we estimate that the plume features have been transported over a distance of 4.3 km to the east-southeast within this time interval."

**16) Figure 10. I am a bit surprised that there are no molecular features visible in the wavelength dependent AMF. (I haven't done the calculation myself – maybe the ozone features would only be visible at shorter wavelengths on this y-axis scale? Just wanted to mention this, to confirm that ozone was properly included in the AMF calculation – if so, ignore this comment.)**

We checked this and ozone was properly included in the AMF calculations. Molecular structures are indeed expected at shorter wavelengths, e.g. in the $SO_2$ fitting window.

We assume $NO_2$ to be optically thin. We expect to see absorption structures by $NO_2$ itself when the concentration is (very) high, e.g. above $5 \times 10^{16}$ molec $cm^{-2}$.

**17) Figure 12, 13, 14: Black symbols, grey road are very hard to see. Consider making larger and thicker. Maybe change road color to white if still hard to see. Wind direction is almost impossible to read in plot, but that is okay as is giving in caption.**

We have updated Fig. 12, 13 and 14 and we have put the black symbols in white and enlarged them. We also changed the A100-A113 highway color in white. Note that in the APEX $NO_2$ VCD map of Fig. 13, we also added two asterisks related to comment 15.

**18) Technical Corrections:**

**P2, Line 10: SCIAMACHY acronym definition is incorrect**

**P 12, Line 21: Change "origin" to "originate"**

OK, SCIAMACHY acronym corrected to "SCanning Imaging Absorption spectroMeter for Atmospheric CHartographY".

[revised manuscript text omitted]